# Reduced expression of C/EBPβ-LIP extends health and lifespan in mice

Christine Müller[1,2†], Laura M Zidek[2†], Tobias Ackermann[1], Tristan de Jong[1], Peng Liu[3], Verena Kliche[2], Mohamad Amr Zaini[1], Gertrud Kortman[1], Liesbeth Harkema[4], Dineke S Verbeek[5], Jan P Tuckermann[3], Julia von Maltzahn[2], Alain de Bruin[4,5], Victor Guryev[1], Zhao-Qi Wang[2], Cornelis F Calkhoven[1,2]*

[1]European Research Institute for the Biology of Ageing, University Medical Centre Groningen, University of Groningen, Groningen, Netherlands; [2]Leibniz Institute on Aging - Fritz Lipmann Institute, Jena, Germany; [3]Institute for Comparative Molecular Endocrinology, University of Ulm, Ulm, Germany; [4]Dutch Molecular Pathology Centre, Faculty of Veterinary Medicine, Utrecht University, Utrecht, Netherlands; [5]Department of Genetics, University Medical Center Groningen, University of Groningen, Groningen, Netherlands

**Abstract** Ageing is associated with physical decline and the development of age-related diseases such as metabolic disorders and cancer. Few conditions are known that attenuate the adverse effects of ageing, including calorie restriction (CR) and reduced signalling through the mechanistic target of rapamycin complex 1 (mTORC1) pathway. Synthesis of the metabolic transcription factor C/EBPβ-LIP is stimulated by mTORC1, which critically depends on a short upstream open reading frame (uORF) in the *Cebpb*-mRNA. Here, we describe that reduced C/EBPβ-LIP expression due to genetic ablation of the uORF delays the development of age-associated phenotypes in mice. Moreover, female C/EBPβ^ΔuORF mice display an extended lifespan. Since LIP levels increase upon aging in wild type mice, our data reveal an important role for C/EBPβ in the aging process and suggest that restriction of LIP expression sustains health and fitness. Thus, therapeutic strategies targeting C/EBPβ-LIP may offer new possibilities to treat age-related diseases and to prolong healthspan.
DOI: https://doi.org/10.7554/eLife.34985.001

*For correspondence:
c.f.calkhoven@umcg.nl

†These authors also contributed equally to this work

Competing interests: The authors declare that no competing interests exist.

## Introduction

Delaying the occurrence of age related-diseases and frailty (disabilities) and thus prolonging health-span, would substantially increase the quality of life of the ageing population and could help to reduce healthcare costs. Calorie restriction (CR) or pharmacological inhibition of the mTORC1 pathway by rapamycin are considered as potential effective interventions to delay aging and to increase healthspan in different species (*Kaeberlein et al., 2015*). However, for humans CR is a difficult practice to maintain and may have pleiotropic effects depending on genetic constitution, environmental factors and stage of life. Likewise, the long-term use of rapamycin is limited by the risk of side effects, including disturbed glucose homeostasis, impaired wound healing, gastrointestinal discomfort and others (*Augustine et al., 2007*; *de Oliveira et al., 2011*; *Lamming et al., 2012*; *Wilkinson et al., 2012*). Therefore, there is a need to investigate alternative targets that are part of the CR/mTORC1 pathway that can be manipulated to reach similar beneficial effects. Our work suggests that the transcription factor C/EBPβ may provide such a target.

C/EBPβ regulates the expression of metabolic genes in liver and adipose tissue (*Desvergne et al., 2006*; *Roesler, 2001*). From its mRNA, three protein isoforms are synthesized through the usage of different translation initiation sites: two isoforms acting as transcriptional

**eLife digest** The risks of major diseases including type II diabetes, cancer and Alzheimer's are linked to the biological process of ageing. By finding ways to slow ageing, we can help more people to live longer healthier lives while avoiding these illnesses.

Placing some animals on a diet that contains only two-thirds as many calories as they would normally eat can improve their fitness during old age and delay the onset of many age-related problems. It is unrealistic to expect people to control their diet to this extent, yet there may be other ways to bring about the same effects.

Calorie restriction affects the activity of many different genes; for example, it causes a gene that produces a protein known as Liver-enriched Inhibitory Protein (LIP for short) to shut down. LIP controls the activity of many genes involved in metabolism, so it could be a key target for drugs to control ageing.

Müller, Zidek et al. used mice that are unable to produce LIP to study this protein's effect on ageing. The life expectancy of female mice lacking LIP increased by up to 20%. These mice were leaner, fitter, more resistant to cancer, had stronger immune systems and controlled their blood sugar levels better than normal mice. Male mice that lacked LIP did not live longer but did experience some ageing-related benefits. Genetic analysis also showed that gene activity particularly of metabolic genes is more robust in old female LIP-deficient mice and thus more similar to young control mice than old control mice.

The results presented by Müller, Zidek et al. suggest that targeting the activity of the LIP gene could help to slow the ageing process. It is not yet clear whether shutting off LIP has similar beneficial effects in humans. Further research is also needed to investigate why female mice gain more benefits from a lack of LIP than males do.

DOI: https://doi.org/10.7554/eLife.34985.002

activators, liver-enriched activator protein (LAP) −1 and −2, and a transcriptional inhibitory isoform called liver-enriched inhibitory protein (LIP) (*Descombes and Schibler, 1991*). We showed earlier that translation into LIP depends on a *cis*-regulatory uORF (*Figure 1A*) and is stimulated by mTORC1 signalling (*Calkhoven et al., 2000*; *Jundt et al., 2005*; *Zidek et al., 2015*). Pharmacological or CR-induced inhibition of mTORC1 in mice selectively reduces LIP-protein synthesis and thereby increases the LAP/LIP ratio in different tissues (*Zidek et al., 2015*). Experimental reduction of LIP expression by genetic ablation of the uORF in C/EBPβ$^{\Delta uORF}$ knockin mice is associated with a CR-type improved metabolic profile, including enhanced fatty acid oxidation and reduction of steatosis, improved insulin sensitivity and glucose tolerance, and higher adiponectin levels. Notably, these metabolic improvements are achieved without reducing calorie intake (*Albert and Hall, 2015*; *Zidek et al., 2015*). Because of the similarities between t h e C/EBPβ$^{\Delta uORF}$ mutation and CR, we investigated lifespan and age-associated phenotypes in C/EBPβ$^{\Delta uORF}$ mice.

Here, we show that the C/EBPβ$^{\Delta uORF}$ mutation is associated with an increase in lifespan and reduced tumour incidence in female mice. In addition, we show an improvement in a broad spectrum of age-associated phenotypes to varying degrees in males and females.

## Results

Others showed that LIP levels increase during aging in liver and white adipose tissue (WAT) (*Hsieh et al., 1998*; *Karagiannides et al., 2001*; *Timchenko et al., 2006*). Similarly, in our cohorts of wt C57BL/6J mice LIP levels are significantly higher in livers of old (20–22 months) versus young (5 months) mice, resulting in a decrease in the LAP/LIP ratio during ageing (*Figure 1B,C* and *Figure 1—figure supplement 1A*). In contrast, in C/EBPβ$^{\Delta uORF}$ mice LIP levels are low and stay low in old mice. LAP levels in C/EBPβ$^{\Delta uORF}$ males and to a lesser extent in females are increased, which is probably due to additional initiation events at the LAP-AUG by ribosomes that normally would have initiated at the uORF (*Calkhoven et al., 2000*). The *Cebpb*-mRNA levels are comparable at different ages and in the different genotypes (*Figure 1D,E*). Similarly, LIP expression is higher in white adipose tissue (WAT) of old female mice (WAT from males is not available) (*Figure 1—figure supplement 1B*). Since translation into LIP is stimulated by mTORC1 through phosphorylation of 4E-binding protein

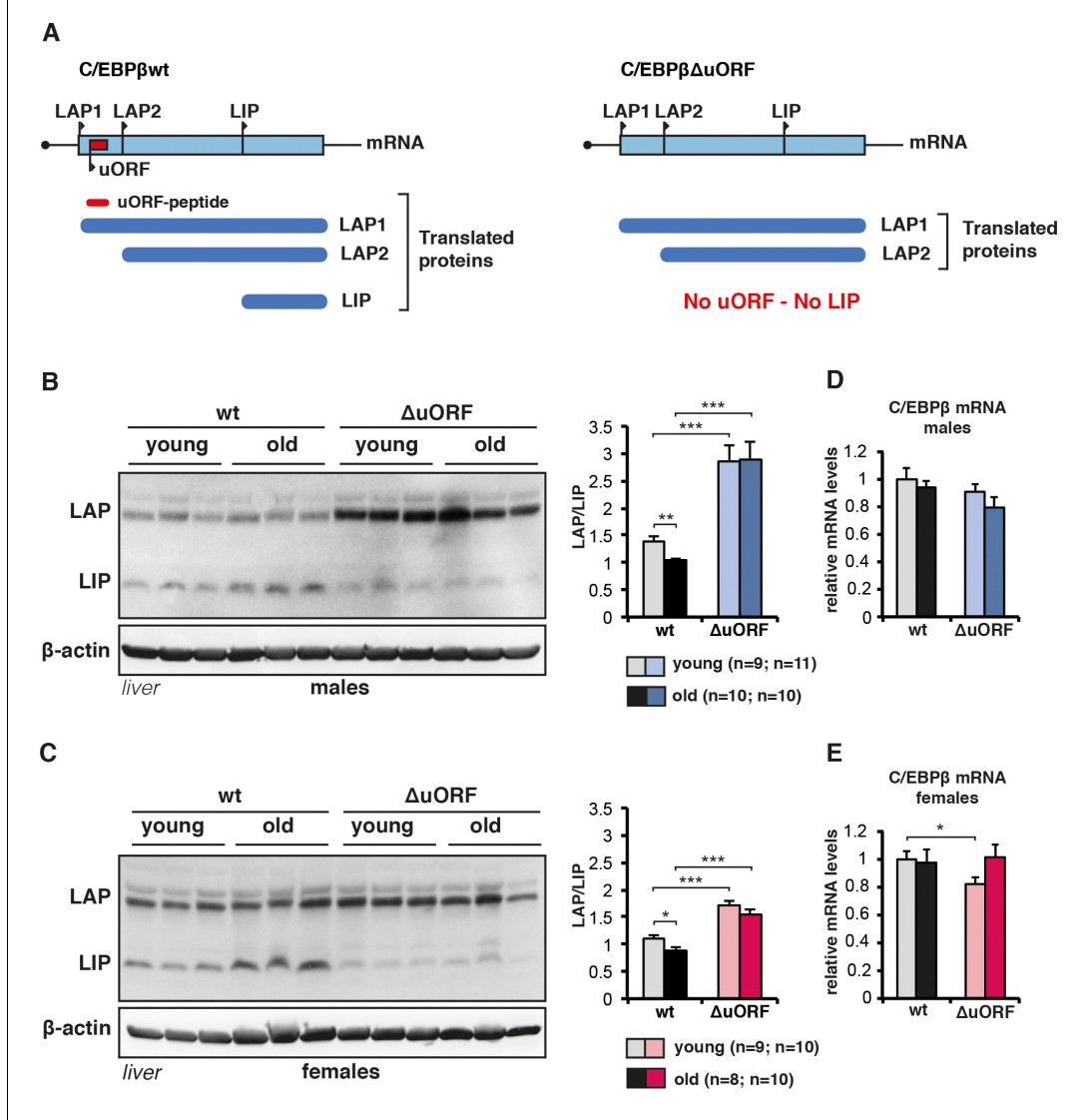

**Figure 1.** C/EBPβ LAP/LIP isoform ratio increases upon ageing. (**A**) The graph at the left shows that wt C/EBPβ-mRNA is translated into LAP1 and LAP2 through regular translation initiation, while translation into LIP involves a primary translation of the uORF followed by translation re-initiation at the downstream LIP-AUG by post-uORF-translation ribosomes. The graph at the right shows that genetic ablation of the uORF abolishes translation into LIP, but leaves translation into LAP1 and LAP2 unaffected (for detailed description see [*Calkhoven et al., 2000*; *Zidek et al., 2015*]). (**B** and **C**) Immunoblots of liver samples from young (5 months) and old (female 20 months, male 22 months) wt and C/EBPβ$^{ΔuORF}$ (**B**) males and (**C**) females showing LAP and LIP isoform expression. β-actin expression served as loading control. The LAP/LIP isoform ratio as calculated from quantification by chemiluminescence digital imaging of immunoblots is shown at the right (wt males n = 9 young, n = 10 old; C/EBPβ$^{ΔuORF}$ males, n = 11 young, n = 10 old; wt females, n = 9 young, n = 8 old; C/EBPβ$^{ΔuORF}$ females, n = 10 young, n = 10 old). (**D** and **E**) C/EBPβ mRNA levels as determined by quantitative real-time PCR in (**D**) males (wt, n = 11 young, n = 11 old; C/EBPβ$^{ΔuORF}$, n = 11 young, n = 9 old) and in (**E**) females (wt, n = 9 young, n = 11 old; C/EBPβ$^{ΔuORF}$, n = 9 young, n = 11 old). P-values were determined by Student's t-test, *p<0.05; **p<0.01; ***p<0.001.
DOI: https://doi.org/10.7554/eLife.34985.003

The following figure supplement is available for figure 1:

**Figure supplement 1.** Analysis of C/EBPβ LAP/LIP isoform ratio and mTORC1 signalling upon ageing.
DOI: https://doi.org/10.7554/eLife.34985.004

(4E-BP) (*Zidek et al., 2015*), we reasoned that the higher LIP levels in aged livers and WAT might correlate with increased mTORC1 signalling with age. While the analysis of mTORC1-downstream phosphorylation of 4E-BP1 a n d p70 ribosomal protein S6 kinase 1 (S6K1) did not reveal a significant difference between young versus old or wt versus C/EBPβ$^{ΔuORF}$ mice in liver (*Figure 1—figure*

supplement 1C,D), 4E-BP1 phosphorylation was significantly higher in old compared to young WAT samples from both wt and C/EBPβ$^{\Delta uORF}$ females (*Figure 1—figure supplement 1E,F*). In contrast phosphorylation of ribosomal S6 protein in WAT was not significantly altered upon ageing. Thus, LIP levels increase with age and this increase is dependent on the uORF in the *Cebpb*-mRNA and seems to correlate with mTORC1/4E-BP1 signalling in WAT but not in the liver.

We hypothesised that the C/EBPβ$^{\Delta uORF}$ mutation may have positive effects on healthspan and lifespan based on the CR-like metabolic improvements in C/EBPβ$^{\Delta uORF}$ mice (*Zidek et al., 2015*). A lifespan experiment was set up comparing C/EBPβ$^{\Delta uORF}$ mice with wt littermates (C57BL/6J) in cohorts of 50 mice of each genotype and gender. The survival curves revealed an increase in median survival of 20.6% (difference in overall survival p=0.0014 log-rank test, n = 50) for the female C/EBPβ$^{\Delta uORF}$ mice compared to wt littermates (*Figure 2A*). From the 10% longest-lived females, nine out of ten were C/EBPβ$^{\Delta uORF}$ mice (*Supplementary file 1*), showing that the maximum lifespan of C/EBPβ$^{\Delta uORF}$ females is significantly increased (p=0.0157 Fisher's exact test). If maximum lifespan is determined by the mean survival of the longest-lived 10% of each cohort, C/EBPβ$^{\Delta uORF}$ females show an increase of 9.14% (p-value=0.00105 Student's t-test.). For the male cohort, we observed a modest increase in median survival of 5.2%, however, the overall survival was not significantly increased (p=0.4647 log-rank test, n = 50) (*Figure 2B*). The increase in median survival of the combined cohort of C/EBPβ$^{\Delta uORF}$ mice (males and females) was 10.5% (with a significant increase in overall survival p=0.0323 log-rank test, n = 100) (*Figure 2—figure supplement 1A* and *supplementary file 1*). The observed median survival for wt females (623 days) is lower than what most other labs have reported for C57BL/6J females. We reasoned that this was due to a high incidence of ulcerative dermatitis (UD) we observed particularly in our female cohort (females: 19 mice or 38% for wt and 26 mice or 52% for C/EBPβ$^{\Delta uORF}$; males: 15 mice or 30% for wt and 10 mice or 20% for C/EBPβ$^{\Delta uORF}$). UD is a common and spontaneous condition in mice with a C57BL/6J background that progress to a severity that euthanasia is inevitable (*Hampton et al., 2012*). Therefore, survival curves were also calculated separately for UD-free mice and for mice that were euthanized because of serious UD (*Figure 2C–F*, *Figure 2—figure supplement 1B,C* and *supplementary file 1* for complete overview). These data show that median lifespan of UD-free wt females i s in a more normal range (740 days) and that the C/EBPβ$^{\Delta uORF}$ mutation results in a significant increase of median survival specifically in females irrespective of the condition of UD. Moreover, the median survival of the C/EBPβ$^{\Delta uORF}$ UD-free females (860.5 days) is higher compared to both wt females and wt males (829 days). The survival curves show an increase in early mortality for the male C/EBPβ$^{\Delta uORF}$ mice in the complete and UD-free cohorts (*Figure 2B,D*). For these cohorts, we performed a daily chi-square test to access differences between wt and C/EBPβ$^{\Delta uORF}$ males on each day of the lifespan and found a significant (p<0.05) reduction in survival only for the UD-free C/EBPβ$^{\Delta uORF}$ males spanning the period 582–637 days, including four mortalities (*Figure 2—figure supplement 1D,E*). Taken together, these data show that a significant lifespan extension can be concluded only for female C/EBPβ$^{\Delta uORF}$ mice.

Aging is the most important risk factor for development of cancer. A reduction in cancer incidence is recurrently observed upon CR, rapamycin-treatment or manipulation of other pathways that increase longevity in several animal models (*Anisimov et al., 2011*; *Colman et al., 2009*; *Komarova et al., 2012*; *Mattison et al., 2012*; *Neff et al., 2013*; *Serrano, 2016*; *Weindruch and Walford, 1982*). Mice in the lifespan cohorts that died or were sacrificed according to humane endpoint criteria underwent necropsy and tumours were analysed by a board certified veterinary pathologists of the Dutch Molecular Pathology Centre (DMPC). The incidence of neoplasms was markedly reduced in female C/EBPβ$^{\Delta uORF}$ mice compared to female wt mice (68% - > 45,8%, p=0.025 Fisher's exact test) (*Figure 3A*). Furthermore, tumours were detected on necropsy at a higher age in female C/EBPβ$^{\Delta uORF}$ mice compared to wt mice indicating a delay in tumour development (*Figure 3C*). The increase in median survival of the tumour bearing C/EBPβ$^{\Delta uORF}$ females was 25.49% compared to that of tumour bearing wt females (p=0.0217 log-rank test) (*Figure 3—figure supplement 1A*). Also the tumour load (number of different tumour types per mouse) and the tumour spread (total number of differently located tumours per mouse irrespective of the tumour type) were lower in female C/EBPβ$^{\Delta uORF}$ mice (*Figure 3—figure supplement 1B*). For males no significant reduction in tumour incidence was detected in C/EBPβ$^{\Delta uORF}$ mice (*Figure 3B,D*). The survival of tumour bearing mice and the tumour load was similar in wt and C/EBPβ$^{\Delta uORF}$ males, while the tumour spread seems to be even slightly increased in C/EBPβ$^{\Delta uORF}$ male mice (*Figure 3—figure supplement 1C,D*). The main

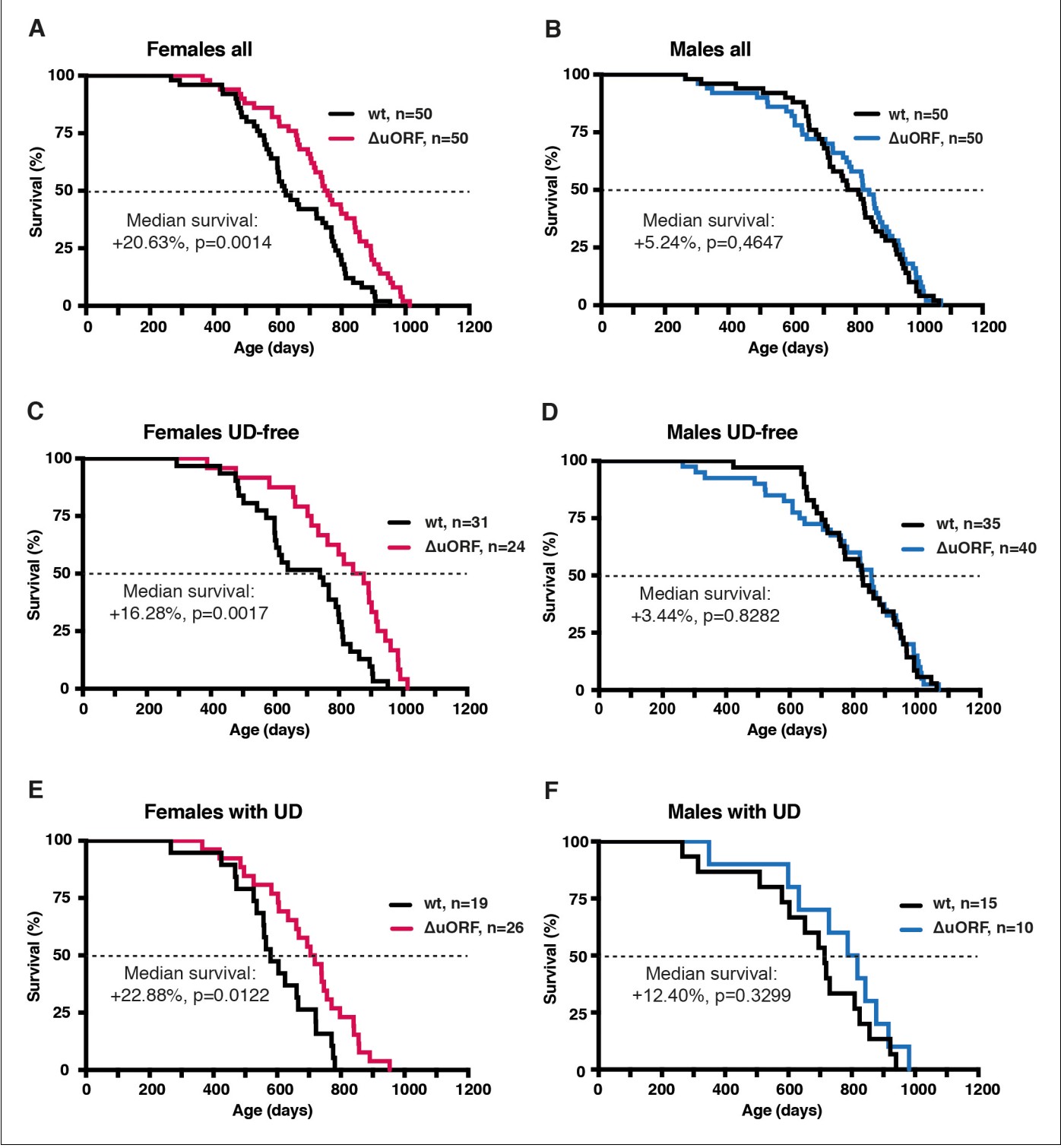

**Figure 2.** Increased survival of female C/EBPβ$^{\Delta uORF}$ mice. Survival curves of (**A**) the complete female cohorts, (**B**) complete male cohorts, (**C**) the UD-free female cohorts, (**D**) UD-free male cohorts, (**E**) female mice with UD and (**F**) male mice with UD with the survival curves of wt or C/EBPβ$^{\Delta uORF}$ mice indicated. The increase in median survival (%) of C/EBPβ$^{\Delta uORF}$ compared to wt littermates and statistical significance of the increase in the overall survival as determined by the log-rank test is indicated in the figure.

DOI: https://doi.org/10.7554/eLife.34985.005

The following figure supplement is available for figure 2:

**Figure supplement 1.** Survival curves of combined male and female cohorts and daily chi-square test for male cohorts.

*Figure 2 continued on next page*

*Figure 2 continued*

DOI: https://doi.org/10.7554/eLife.34985.006

tumour types found in female mice were lymphoma, hepatocellular carcinoma and histiocytic sarcoma. The occurrence of all three types was reduced in C/EBPβ$^{\Delta uORF}$ females (*Supplementary file 2*). For other tumour types, the single numbers are too small to make a clear statement about a change in frequency. In male mice, hepatocellular carcinoma and histiocytic sarcoma were the most frequent tumour types observed. Although the overall tumour incidence was similar in C/EBPβ$^{\Delta uORF}$ and wt males, the frequency of hepatocellular carcinoma was reduced in the C/EBPβ$^{\Delta uORF}$ males (*Supplementary file 2*).

Apart from the reduced tumour incidence and the increase in survival of tumour-bearing C/EBPβ$^{\Delta uORF}$ females, also the survival of tumour-free female C/EBPβ$^{\Delta uORF}$ mice was significantly extended by 25.13% (p=0.0467 log-rank test) compared to wt tumour-free females (*Figure 3—figure supplement 1E*). This suggests that both the tumour incidence and additional unrelated factors contribute to the increased survival of C/EBPβ$^{\Delta uORF}$ females. The observed increase in median lifespan of tumour-free C/EBPβ$^{\Delta uORF}$ males of 19.71% does not correlate with a statistically significant

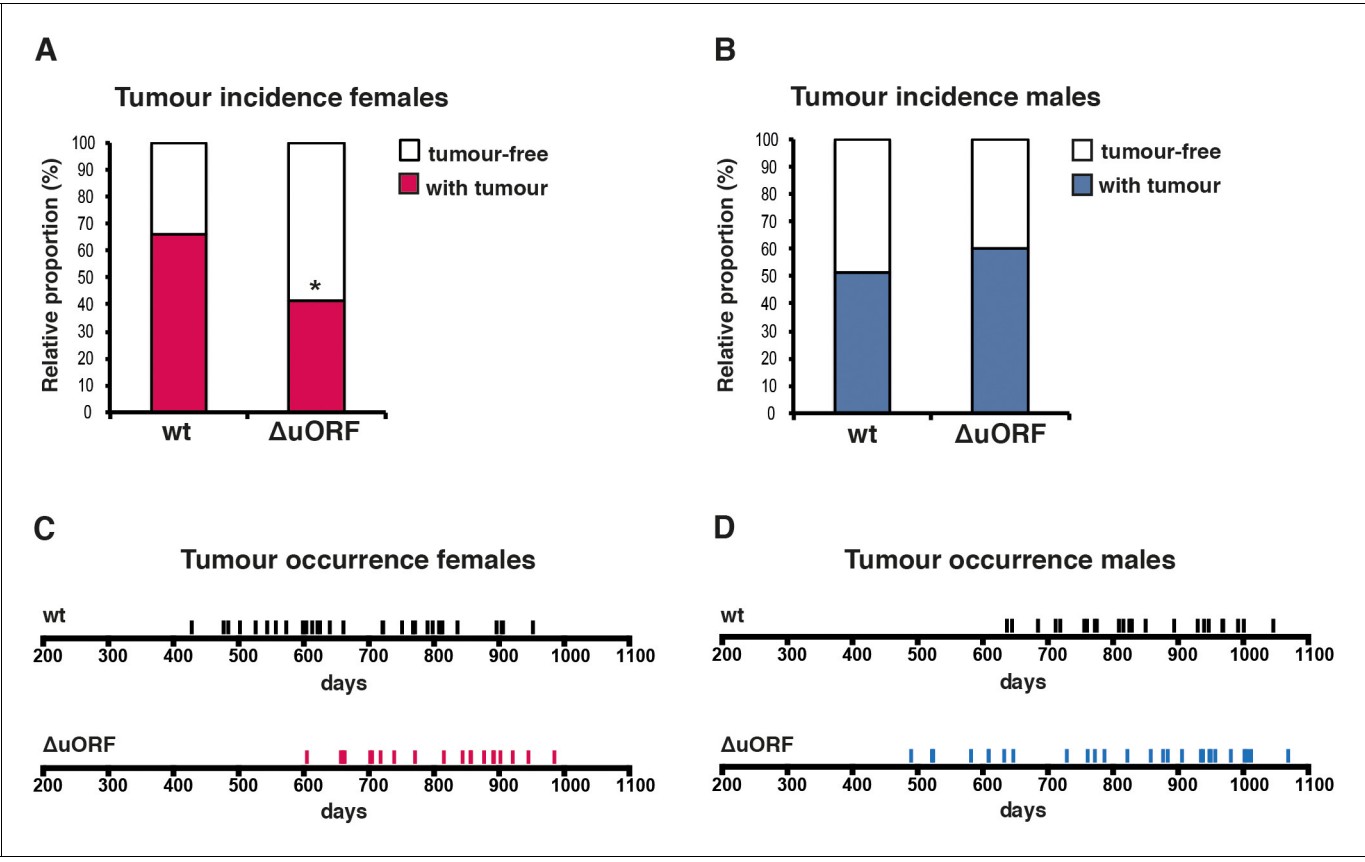

**Figure 3.** Reduced incidence and delayed occurrence of tumours in female C/EBPβ$^{\Delta uORF}$ mice. (**A**) Tumour incidence of females as determined by pathological examination of neoplasms found upon necropsy of mice from the lifespan cohorts (wt, n = 50; C/EBPβ$^{\Delta uORF}$, n = 48). Statistical significance was calculated using Fisher's exact test with *p<0.05. (**B**) Tumour incidence of males as determined by pathological examination of neoplasms found upon necropsy (wt, n = 47; C/EBPβ$^{\Delta uORF}$, n = 45). (**C**) Tumour occurrence in the female lifespan cohorts upon necropsy is shown for wt (black lines) and C/EBPβ$^{\Delta uORF}$ mice (red lines). (**D**) Tumour occurrence in the male lifespan cohorts upon necropsy is shown for wt (black lines) and C/EBPβ$^{\Delta uORF}$ mice (blue lines).

DOI: https://doi.org/10.7554/eLife.34985.007

The following figure supplement is available for figure 3:

**Figure supplement 1.** Analysis of tumour related survival, and tumour load and spread.

DOI: https://doi.org/10.7554/eLife.34985.008

increase in the overall survival (p=0.4647 log-rank test) (*Figure 3—figure supplement 1F*). However, the survival curve points to a possible health improvement in the median phase of the male lifespan. Taken together, the C/EBPβ$^{\Delta uORF}$ mutation in mice restricting the expression of LIP results in a significant lifespan extension and decreased tumour incidence in females but not in males.

Typically, CR-mediated, genetic or pharmacological suppression of mTORC1 signalling is accompanied by the attenuation of an age-associated decline of health parameters (*Johnson et al., 2013*). We examined the selected health parameters of body weight and composition, glucose tolerance, naïve/memory T-cell ratio, motor coordination and muscle strength in separate ageing cohorts of young (3–5 months) and old (18–20 months for females and 20–22 months for males) mice. In addition, we compared the histological appearance of selected tissues (liver, muscle, pancreas, skin, spleen and bone) between old (20/22 months) wt and C/EBPβ$^{\Delta uORF}$ mice. Body weight was significantly increased in all old mice (*Figure 4A,B*). The increase for the old female C/EBPβ$^{\Delta uORF}$ mice was significantly smaller compared to old wt littermates, while for the males there was no significant difference between the genotypes (*Figure 4A,B*). The slightly lower body weight for the young C/EBPβ$^{\Delta uORF}$ males was also observed in our previous study (*Zidek et al., 2015*). A similar pattern was observed regarding the fat content that was measured by abdominal computed tomography (CT) analysis (*Figure 4C,D* and *Figure 4—figure supplement 1C*). The volumes of total fat increased strongly in old mice both in visceral and subcutaneous fat depots (*Figure 4—figure supplement 1A, B*). Old female C/EBPβ$^{\Delta uORF}$ mice accumulated significantly less fat in the visceral and subcutaneous fat depots than wt females, while there was no difference for male mice (*Figure 4—figure supplement 1A,B*). The lean body mass was slightly lower in old female C/EBPβ$^{\Delta uORF}$ mice and increased in male wt mice compared to young mice (*Figure 4—figure supplement 1A,B*). Thus, female C/EBPβ$^{\Delta uORF}$ mice gain less fat upon aging similar to mice under CR or upon prolonged rapamycin treatment (*Fang et al., 2013*). In contrast, although male C/EBPβ$^{\Delta uORF}$ mice had a lower body weight and subcutaneous fat content at a young age compared to wt mice they were not able to maintain this difference during the aging process, which correlates with the lack in lifespan extension. In addition, we found an increase in mRNA expression of the macrophage marker *Cd68* as a measure for age-related macrophage infiltration in visceral WAT of old mice, which was attenuated in female but not in male C/EBPβ$^{\Delta uORF}$ mice (*Figure 4—figure supplement 1D*).

Impaired glucose tolerance is a hallmark of the aging process, which is improved by CR (*Barzilai et al., 1998*; *Mitchell et al., 2016*). The intraperitoneal glucose tolerance test (IPGTT) showed that glucose clearance, calculated as the area under the curve (AUC), is significantly less efficient in old wt compared to young wt mice (*Figure 4E,F*). Old C/EBPβ$^{\Delta uORF}$ females and males perform significantly better in the IPGTT test than old wt littermates, which is reflected by the lower AUC value. Therefore, the C/EBPβ$^{\Delta uORF}$ mutation protects against age-related decline of glucose tolerance in males and females.

The ageing associated increase in memory/naïve T-cell ratio is a robust indicator for the progression of the immunological ageing progress. At a young age naïve T cells predominate and memory T cells are relatively scarce. Upon ageing the naïve T cell population is strongly reduced with a concomitant increase in the memory T cell population, resulting in an increased ratio of memory to naïve T cells (*Hakim et al., 2004*). The ratio of memory (Cd44high) to naïve (Cd44low/Cd62Lhigh) cytotoxic T (Cd8+) cells or memory (Cd44high) to naïve (Cd44low/Cd62Lhigh) helper T (Cd4+) cells was analysed by flow cytometric analysis. Both increased upon aging in the blood of males and females of both genotypes (*Figure 5A–D*). However, in C/EBPβ$^{\Delta uORF}$ mice of both genders, this increase was significantly attenuated compared to wt mice (*Figure 5A–D* and *Figure 5—figure supplement 1A–D*). These data suggest that the C/EBPβ$^{\Delta uORF}$ mutation preserves a more juvenile immunological phenotype during ageing.

Aging is associated with a significant decline in motor coordination and muscle strength (*Barreto et al., 2010*; *Demontis et al., 2013*). In the rotarod test, the time is measured that mice endure on a turning and accelerating rod as an indication for their motor-coordination. As expected, rotarod performance decreased with age both for wt female and male mice (*Figure 6A*). Remarkably, rotarod performance was completely preserved in old C/EBPβ$^{\Delta uORF}$ females but not in C/EBPβ$^{\Delta uORF}$ males. In the beam walking test, the required crossing time and number of paw slips of mice traversing a narrow beam are measured. Old mice needed more time to cross the beam reflecting loss of motor coordination upon ageing (*Figure 6B*). The aging-associated increase of the crossing time was less severe in C/EBPβ$^{\Delta uORF}$ males and females, although statistically significant

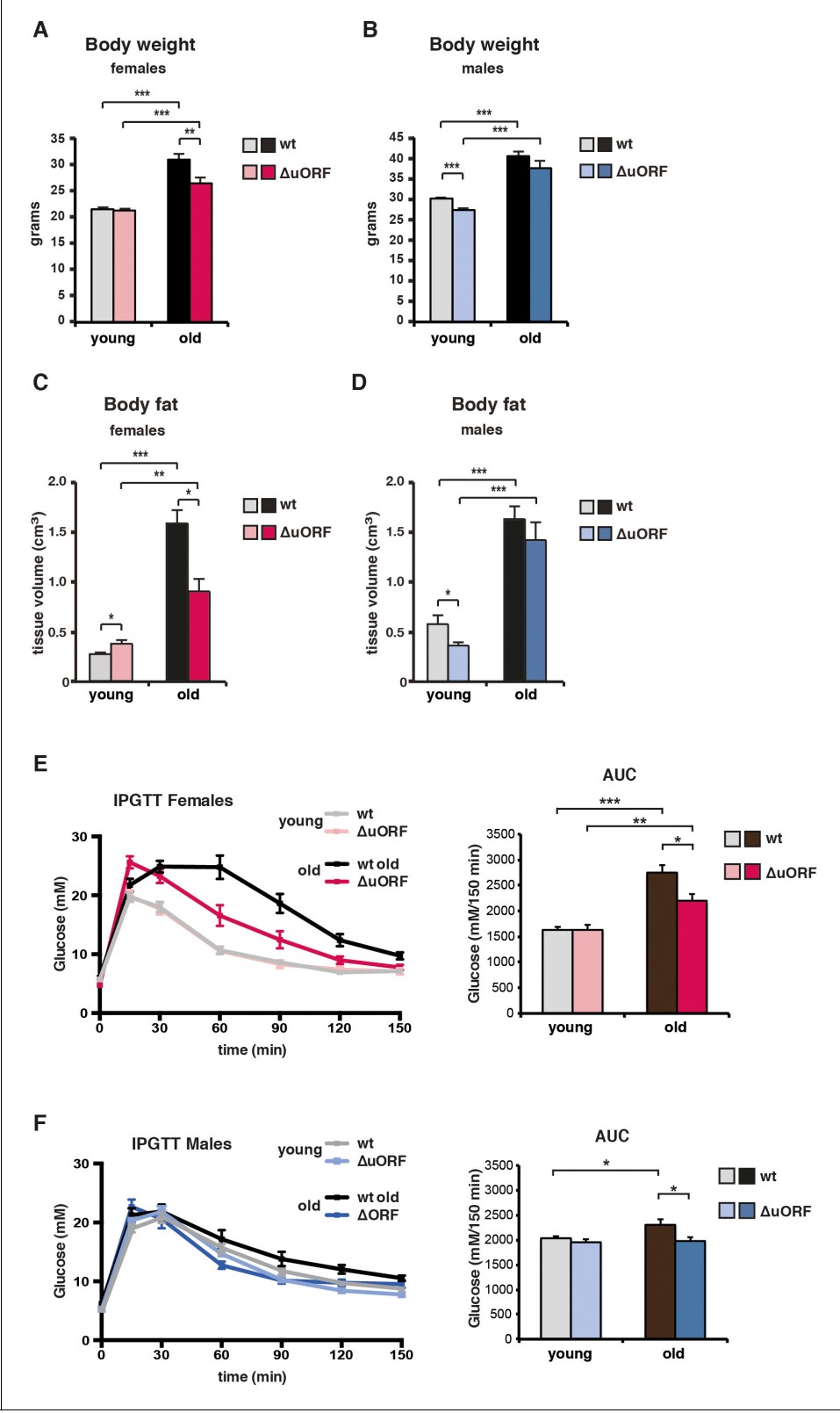

**Figure 4.** Ageing-associated increase in body weight, fat content and glucose tolerance is attenuated in female C/EBPβ[ΔuORF] mice. (**A**) Body weight (g) of young (4 months) and old (19 months) female mice (wt, n = 11 young, n = 12 old; C/EBPβ[ΔuORF], n = 11 young, n = 12 old). (**B**) Body weight of young (4 months) and old (21 months) male mice (wt, n = 12 young and old; C/EBPβ[ΔuORF], n = 12 young, n = 11 old). (**C**) Body fat content (cm³) as determined by CT analysis of young (4 months) and old (19 months) female mice (wt, n = 11 young and old; C/EBPβ[ΔuORF], n = 9 young, n = 11 old). (**D**) Body fat

*Figure 4 continued on next page*

Figure 4 continued

content of young (4 months) and old (21 months) male mice (wt, n = 12 young and old wt; C/EBPβ^ΔuORF, n = 11 young, n = 9 old). (**E** and **F**) i.p.-Glucose Tolerance Test (IPGTT) was performed with young (4 months) and old (female 19 months, male 21 months) wt and C/EBPβ^ΔuORF (**E**) females and (**F**) males. The area under the curve (AUC) at the right shows the quantification (wt females, n = 9 young, n = 10 old; C/EBPβ^ΔuORF females, n = 10 young, n = 11 old; wt males, n = 12 young, n = 11 old; C/EBPβ^ΔuORF males, n = 11 young, n = 10 old). P-values were determined by Student's t-test, *p<0.05; **p<0.01; ***p<0.001.

DOI: https://doi.org/10.7554/eLife.34985.009

The following figure supplement is available for figure 4:

**Figure supplement 1.** Analysis of lean body mass, fat volume, and expression of theCd68macrophage marker in young and old mice.
DOI: https://doi.org/10.7554/eLife.34985.010

only in males (*Figure 6B*). Nevertheless, the strong increase in the number of paw slips in old wt mice is almost completely attenuated in C/EBPβ^ΔuORF males and females (*Figure 6C*). Note that the number of paw slips by young C/EBPβ^ΔuORF males is already significantly lower compared to young wt males. During the wire hang test, the time is measured that mice endure hangi n g from an elevated wire which serves as an indication for limb skeletal muscle strength (*Brooks and Dunnett, 2009*). Similar to the rotarod test, the decline in wire hang performance that is seen in old wt mice is completely restored for the female but not for the male C/EBPβ^ΔuORF mice (*Figure 6D*).

Taken together, these data demonstrate that the decline in motor coordination and muscle strength is less severe and partly abrogated in female C/EBPβ^ΔuORF mice. The results for the old male C/EBPβ^ΔuORF mice are not that clear since they show an improved performance only in the beam walking test. One possible explanation is that only the beam walking test measures purely motor coordination skills whereas the results from the rotarod and wire hang tests are influenced in addition by muscle strength and endurance. Old C/EBPβ^ΔuORF males thus might have maintained their motor coordination upon ageing but still suffer from an ageing-dependent loss of muscle strength.

By histological examination of different tissues, we observed a reduction in some age-related alterations in C/EBPβ^ΔuORF mice compared to old wt controls (*Supplementary file 3*). We observed a reduced severity of hepatocellular vacuolation and cytoplasmic nuclear inclusions in male C/EBPβ^ΔuORF mice; in the pancreas both male and female C/EBPβ^ΔuORF mice showed a reduced occurrence and severity of islet cell hyperplasia; in skeletal muscle the number of regenerating muscle fibres was higher in male C/EBPβ^ΔuORF mice; the incidence of dermal inflammation was lower in female C/EBPβ^ΔuORF mice. Unexpectedly, a slightly increased level of inflammation was detected in the livers of female C/EBPβ^ΔuORF mice. The incidence of other potential age-related pathologies like focal acinar cell atrophy and inflammation in the pancreas, liver polyploidy, spleen lymphoid hyperplasia and extramedullary haematopoiesis, intramuscular adipose tissue infiltration, subcutaneous fat atrophy and bone density were not significantly altered between old wt and C/EBPβ^ΔuORF mice. We found slightly reduced plasma IGF-1 levels in old C/EBPβ^ΔuORF females compared to old wt females (*Supplementary file 3*). A reduction in circulating IGF-1 levels was also found in mice under CR and is believed to be an important mediator of health- and lifespan extending effects of CR (*Breese et al., 1991*; *Mitchell et al., 2016*). Taken together, our data show that multiple, but not all, ageing-associated alterations are attenuated in C/EBPβ^ΔuORF mice, and to different extends in males and females.

Finally, we performed a comparative transcriptome analysis from livers of 5 and 20 months old wt and C/EBPβ^ΔuORF female mice (*de Jong and Guryev, 2018a*; *de Jong and Guryev, 2018b*; *Müller et al., 2018*). A principal component analysis revealed that there was a clear effect of the genotype on gene expression only in the old mice suggesting that the differences in gene expression between wt and C/EBPβ^ΔuORF mice are aging dependent (*Figure 7—figure supplement 1*). This is supported by the finding that in young mice only 42 genes were differentially regulated between wt and C/EBPβ^ΔuORF mice (FDR < 0.01; 24 genes upregulated and 18 genes down-regulated in C/EBPβ^ΔuORF mice compared to wt mice) while in old mice we found 152 differentially regulated genes (FDR < 0.01; 127 genes upregulated and 25 genes downregulated in C/EBPβ^ΔuORF mice compared to wt mice). Gene ontology (GO) analysis using the David database (*Huang et al., 2009*) of the genes upregulated in old C/EBPβ^ΔuORF mice in comparison to old wt mice revealed GO terms including 'External side of plasma membrane', 'Positive regulation of T-cell proliferation', and

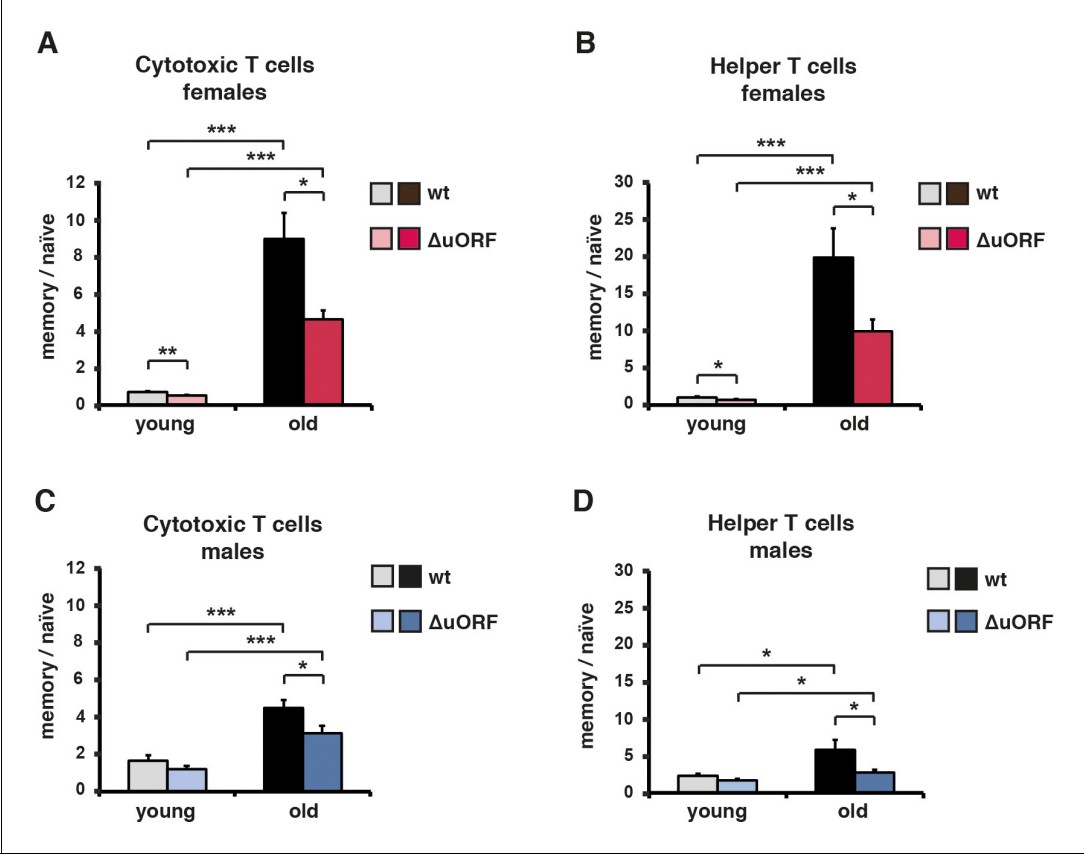

**Figure 5.** Ageing-associated increase of the memory/naïve T-cell ratio is attenuated in C/EBPβ$^{ΔuORF}$ mice. The ratio between Cd44$^{high}$ memory T cells and Cd44$^{low}$/Cd62L$^{high}$ naïve T cells in blood is shown for young (5 months) and old (female 20 months, male 22 months) (**A, B**) females and (**C, D**) males for both (**A, C**) Cd8$^+$ cytotoxic and (**B, D**) Cd4$^+$ helper T cells as was determined by flow cytometry (wt females, n = 10 young, n = 12 old; wt males n = 12 young and old; C/EBPβ$^{ΔuORF}$ females, n = 10 young, n = 12 old; C/EBPβ$^{ΔuORF}$ males, n = 12 young and old). P-values were determined by Student's t-test, *p<0.05; ***p<0.001.

DOI: https://doi.org/10.7554/eLife.34985.011

The following figure supplement is available for figure 5:

**Figure supplement 1.** Analysis of memory / naïve T-cell ratios of cytotoxic and helper T-cells in young and old mice.

DOI: https://doi.org/10.7554/eLife.34985.012

'immune response' (see *Supplementary file 4* for the complete list of GO-terms) whereas the GO-terms: 'Acute phase' and 'Extracellular space' were significantly downregulated (*Supplementary file 5*). Despite the improved metabolic phenotype of C/EBPβ$^{ΔuORF}$ mice (*Zidek et al., 2015*), the analysis did not reveal GO-terms related to metabolism. We reasoned that metabolic genes might not be detected as differentially regulated because they are subject of expression heterogeneity in old mice. Comparison between the coefficient of variation of individual transcripts between young and old mice revealed that inter-individual variation of gene expression increases with age in both genotypes (*Figure 7A,B*) supporting earlier observations made by others (*White et al., 2015*). Direct comparison between old wt and C/EBPβ$^{ΔuORF}$ mice showed that this effect is less pronounced in C/EBPβ$^{ΔuORF}$ mice (*Figure 7C*). KEGG (Kyoto Encyclopedia of Genes and Genomes) pathway and GO-term enrichment analysis of the highly variably expressed genes in the aged livers revealed that in wt mice particularly metabolic genes related to fatty acid metabolism and oxidative phosphorylation were affected which was not observed in C/EBPβ$^{ΔuORF}$ mice (*Figure 7D* and *supplementary file 6* and *7*). In addition, genes whose de-regulation is connected to ageing-associated diseases like non-alcoholic fatty liver disease, Alzheimer's disease, Parkinson's disease, Huntington's disease and cancer were affected by high inter-individual variation in expression levels in old wt but not in old C/EBPβ$^{ΔuORF}$ mice (*Figure 7D*). On the other hand, genes involved in cell cycle, transcription and RNA biology showed higher inter-individual variation in old C/EBPβ$^{ΔuORF}$ mice compared to wt controls

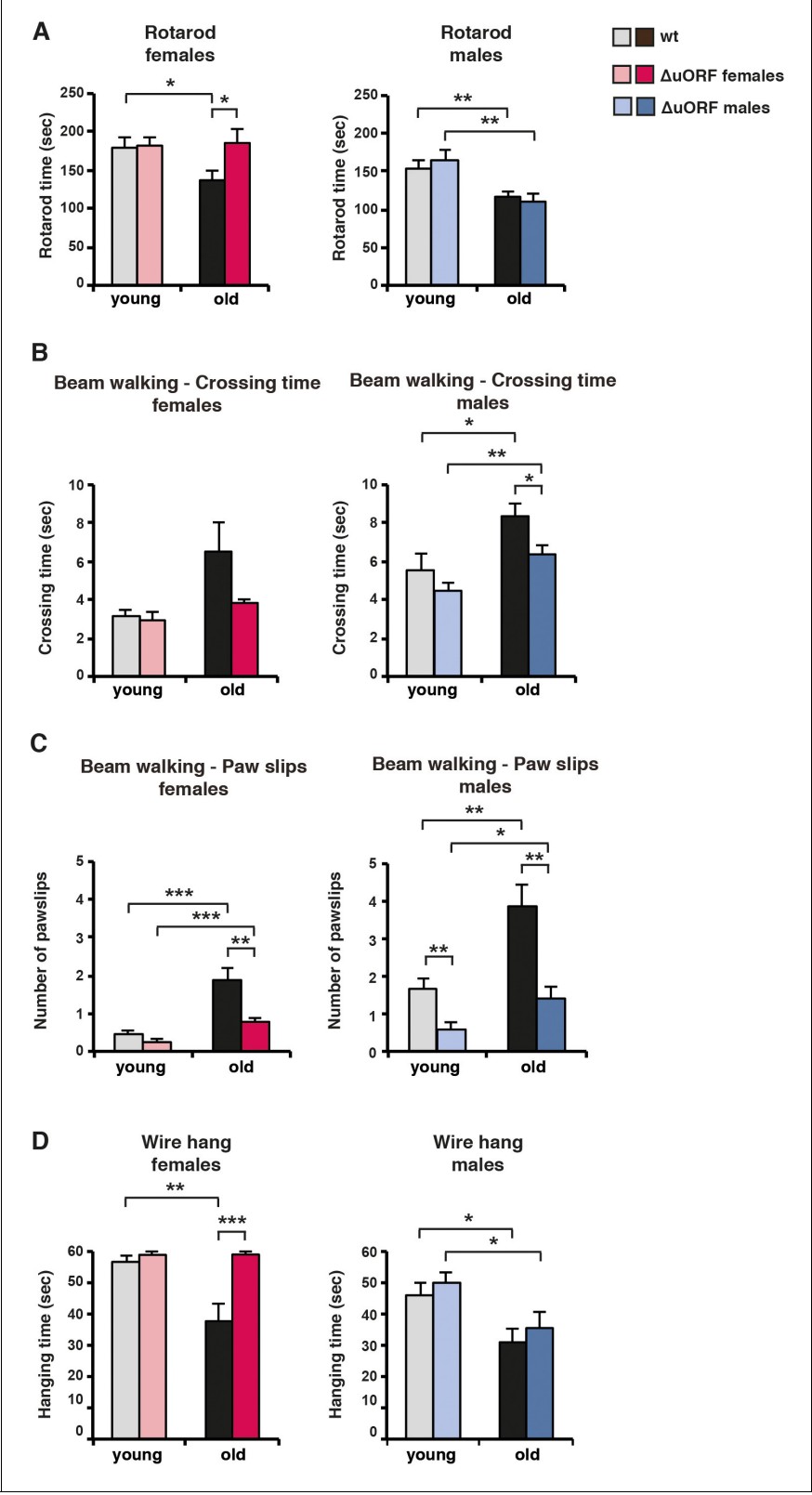

**Figure 6.** Ageing-associated loss of motor coordination and grip strength is attenuated in C/EBPβ$^{\Delta uORF}$ mice. (**A**) Rotarod performance (time in sec of stay on the rotarod) of young (4 months) and old (female 19 months, male 21 months) wt and C/EBPβ$^{\Delta uORF}$ mice is shown separately for females (left) and males (right) (wt females, n = 11 young, n = 12 old; wt males, n = 12 young and old; C/EBPβ$^{\Delta uORF}$ females, n = 11 young, n = 12 old; C/EBPβ$^{\Delta uORF}$ males, n = 12 young, n = 11 old). (**B**) The crossing time (sec) of the beam walking test of young and old wt and C/EBPβ$^{\Delta uORF}$ mice, and (**C**) the number

*Figure 6 continued on next page*

*Figure 6 continued*

of mistakes (paw slips) made while crossing the beam is shown separately for females and males (wt females, n = 11 young, n = 12 old; wt males, n = 12 young and old; C/EBPβ$^{\Delta uORF}$ females, n = 11 young, n = 12 old; C/EBPβ$^{\Delta uORF}$ males, n = 12 young, n = 10 old). (D) Grip strength as determined with the wire hang test as hanging time (sec) of young and old wt and C/EBPβ$^{\Delta uORF}$ mice for females and males separately. N = 11 for young wt and C/EBPβ$^{\Delta uORF}$ females and for old C/EBPβ$^{\Delta uORF}$ females and n = 12 for old wt females; n = 11 for young and old wt males; n = 12 for young C/EBPβ$^{\Delta uORF}$ males and n = 10 for old C/EBPβ$^{\Delta uORF}$ males. P-values were determined by Student's t-test, *p<0.05; **p<0.01; ***p<0.001.

DOI: https://doi.org/10.7554/eLife.34985.013

(*Supplementary file 7*). These findings suggest that expression control of metabolic genes and genes involved in ageing-associated diseases stays more robust upon aging in C/EBPβ$^{\Delta uORF}$ mice.

## Discussion

Taken together, here we show that loss-of-function mutation of a single *cis*-regulatory element - the uORF - in the *Cebpb*-mRNA, which prevents the translation into the transcription factor C/EBPβ-LIP, results in a remarkable juvenile phenotype in aged mice including lower cancer incidence, lower body weight and body fat, better glucose tolerance, lower memory/naïve T cell ratios, and better maintenance of motor coordination. However, we observed clear differences between males and females, with only females showing improvements for cancer incidence, body weight, fat content, Rotarod- and wire hang test performance. In addition, a significant lifespan extension was only observed for the female C/EBPβ$^{\Delta uORF}$ mice.

We do not know what causes the female specific lifespan extension. C/EBP transcription factors are known for their crosstalk with hormone receptors, including estrogen, progesterone and glucocorticoid receptors (*Calkhoven et al., 1997*; *Chang et al., 2005*; *Grøntved et al., 2013*; *Rotinen et al., 2009*; *Seagroves et al., 2000*; *Siersbæk et al., 2012*; *Zhang et al., 2010*). Therefore, obvious differences in hormone receptor regulation between males and females may determine t h e outcome of shifts in LAP/LIP ratios. Notably, the C/EBPβ$^{\Delta uORF}$ mutation in males results in higher LAP expression in the liver and therefore 1.5 fold higher LAP/LIP ratios compared to females (*Figure 1B,C*). Possibly, higher LAP levels in males have some adverse effects on health and lifespan, which may neutralize the beneficial effects of LIP deficiency. In line with this assumption is that the C/EBPβ$^{\Delta uORF}$ males show an increase in early deaths (*Figure 2B,D*) that is significant in UD-free males (*Figure 2—figure supplement 1E*) and is mainly due to early cancer development (*Figure 3D*). A similar scenario has been described for short-term treatment with a high dose of rapamycin that failed to extend t h e lifespan of female mice due to frequent development of aggressive haematological cancers (*Bitto et al., 2016*).

The sex-dependent differences we found are intriguing in the light of studies investigating the lifespan extending effects of CR, rapamycin, and mutations in the mTORC1 pathway. For example, CR by 20% has a greater lifespan extending effect in female C57BL/6J or DBA/2J mice compared to males (*Mitchell et al., 2016*). In addition, moderate overexpression of the mTORC1-upstream inhibitor TSC1 or deletion of the mTORC1-downstream S6K1 results in lifespan extension only in females (*Selman et al., 2009*; *Zhang et al., 2017*). Notably, downregulation of LIP under low mTORC1 signalling is dependent on 4E-BP1/2 function and not on inhibition of S6K1 (*Zidek et al., 2015*). Thus, the bias towards female lifespan extension upon reduced mTORC1 signalling seems to be a common feature irrespective of whether the S6K1 or 4E-BP branch is affected. Mutations affecting both mTORC1 and mTORC2 show ambiguous effects; lifespan extension is limited to females in mice heterozygous for mTOR and its cofactor mammalian lethal with Sec 13 protein 8 (mLST8) (*Lamming et al., 2012*), while in a mTOR-hypomorphic mouse model lifespan extension is observed in both males and females (*Wu et al., 2013*). Similarly, inhibition of mTORC1 with rapamycin results in either a gender biased or a general lifespan extension depending on the study design and rapamycin concentration used. For example, treatment of genetically heterogeneous mice as well as C57BL/6J or C57BL/6Nia mice with a low dose of rapamycin (from 4.7 to 14 ppm) for different time periods has lifespan extending effects that are stronger in females than in males (*Fok et al., 2014*; *Harrison et al., 2009*; *Miller et al., 2011*; *Miller et al., 2014*; *Zhang et al., 2014*). In contrast, treatment with higher concentrations of rapamycin (42 ppm) results in a further increase in lifespan and almost completely alleviates the difference between the sexes (*Miller et al., 2014*). However,

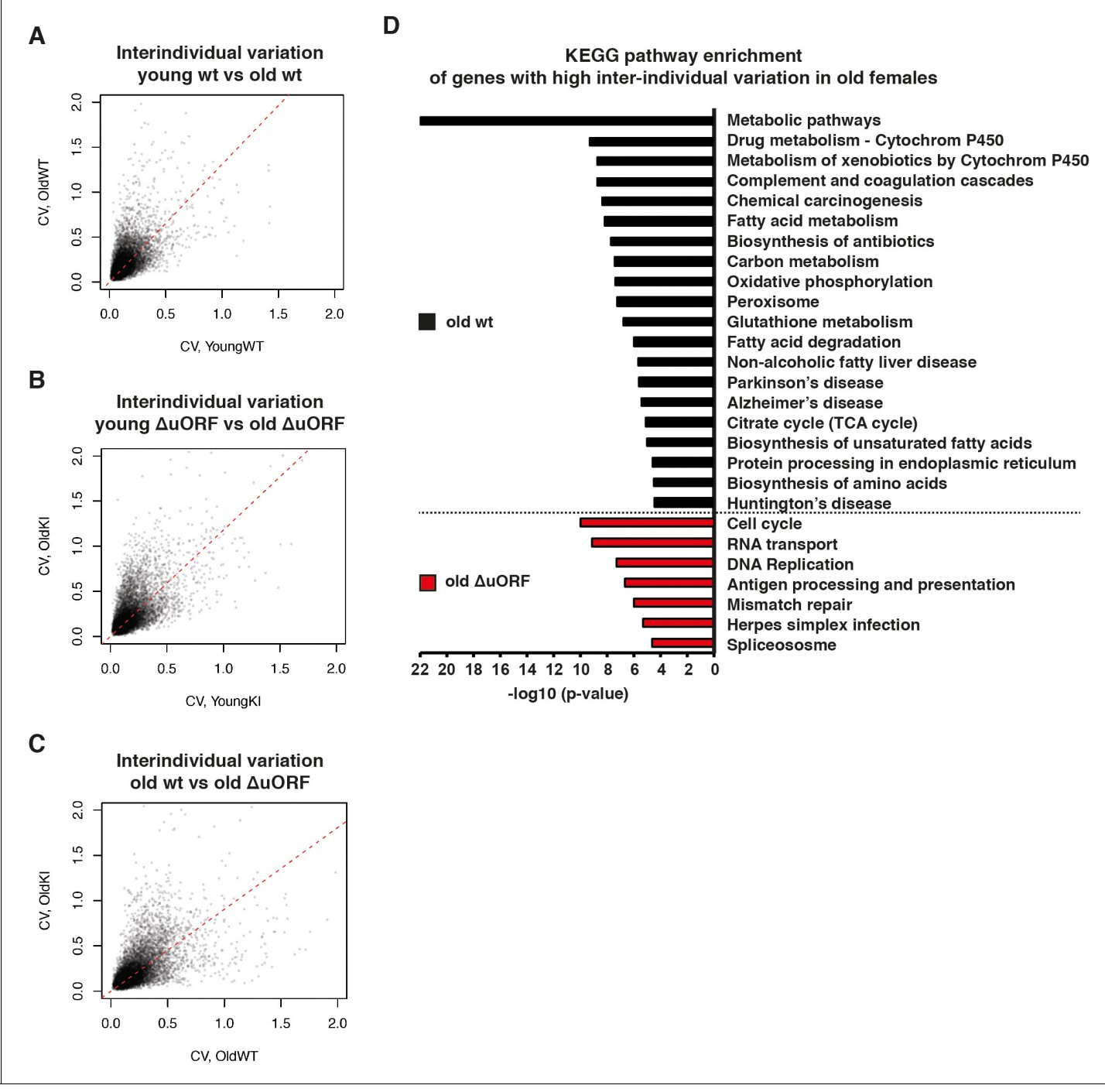

**Figure 7.** Ageing-associated increase of inter-individual variation of gene expression affects different genes in livers from wt and C/EBPβ$^{ΔuORF}$ mice. (A–C) Inter-individual variability of liver transcripts compared between (A) young (5 months) versus old (20 months) female wt mice, (B) young (5 months) versus old (20 months) C/EBPβ$^{ΔuORF}$ female mice and (C) old wt (20 months) versus old C/EBPβ$^{ΔuORF}$ (20 months) female mice (n = 6 for young and old wt and C/EBPβ$^{ΔuORF}$ for A,B,C). Coefficient of variation of transcripts with mean expression >1 FPM is plotted against the coefficient of variation of the other group as indicated. Dashed red line represents linear regression and is shifted towards the side that shows higher inter-individual variability. (D) KEGG pathway enrichment analysis of genes that show increased inter-individual variability in livers from old wt females compared to old C/EBPβΔuORF females (Coefficient of variation of wt genes is more than twice as the coefficient of variation of the same gene in C/EBPβ$^{ΔuORF}$ females) as indicated by the black bars or of genes that show increased inter-individual variability in livers from old C/EBPβ$^{ΔuORF}$ females compared to old wt C/EBPβ$^{ΔuORF}$ females (Coefficient of variation of C/EBPβ$^{ΔuORF}$ genes is more than twice as the coefficient of variation of the same gene in wt females) as indicated by the red bars. The x-axis indicates the p-value. Only pathways that show significant enrichment (FDR < 0.05) are shown.

*Figure 7 continued on next page*

*Figure 7 continued*

DOI: https://doi.org/10.7554/eLife.34985.014

The following figure supplement is available for figure 7:

**Figure supplement 1.** Principal component analysis (PCA) of transcriptome profiles.

DOI: https://doi.org/10.7554/eLife.34985.015

injection of an even higher rapamycin dose (8 mg/kg/day, corresponding to 378 ppm dietary rapamycin) extended lifespan only in males and not in females with serious side effects in females as mentioned above (*Bitto et al., 2016*). These data indicate that rapamycin treatment with low and probably sub-optimal doses creates differences between sexes (*Kaeberlein, 2014*). Although the mechanisms behind these sex-dependent differences are not known, our study suggests that mTORC1-LIP regulation may be involved. Possibly, lifespan-extending pathways downstream of mTORC1 are differentially affected by different rapamycin concentrations, and in a gender dependent way. Providing LIP expression is downregulated by low concentrations of rapamycin the female-biased effect on lifespan might be determined predominantly by low LIP levels as well as by the regulation of other highly sensitive targets like for example S6K1 that similarly shows female-specific effects (*Selman et al., 2009*). At higher rapamycin doses, additional pathways might be engaged from which both males and females benefit. Finally, at too high rapamycin concentrations additional adverse (gender specific) effects might counteract the beneficial effects of rapamycin. Therefore, further research on both positive and negative events downstream of mTORC1 is required to be able to tailor treatment and to minimalize side effects.

Also in mouse strains with alterations in other pathways like the somatotropic axis lifespan extension is often, but not always, more pronounced in females (*Brown-Borg, 2009*). Examples of somatotropic-related female biased lifespan extension are Ames dwarf mice that are deficient in growth hormone (GH) and prolactin production (*Brown-Borg et al., 1996*) and insulin-like growth factor 1 (IGF-1) receptor heterozygous mice (*Holzenberger et al., 2003*). Also in these mouse models the reason for the female-biased lifespan extension is not known.

What contributes to the extended lifespan in the female C/EBPβ$^{ΔuORF}$ mice? Our data indicate that reduced tumour incidence is involved. In line with this is that knockin mice with elevated LIP levels show an increased tumour incidence upon ageing that goes along with reduced survival compared to wt controls (*Bégay et al., 2015*). LIP overexpression can stimulate cell proliferation, migration and transformation in vitro and high LIP levels have been detected in different human tumour tissues (*Anand et al., 2014*; *Arnal-Estapé et al., 2010*; *Calkhoven et al., 2000*; *Haas et al., 2010*; *Jundt et al., 2005*; *Park et al., 2013*; *Raught et al., 1996*; *Zahnow et al., 1997*). Together, these studies suggest an oncogenic role of LIP and that the reduction of LIP in the C/EBPβ$^{ΔuORF}$ mice counteracts tumour development at least partially by cell intrinsic mechanisms. Although the incidence of certain tumours like hepatocellular carcinoma is similarly reduced in male C/EBPβ$^{ΔuORF}$ mice (*Supplementary file 2*) the overall tumour incidence was not different in comparison to the wt males, again indicating gender specific effects of the C/EBPβ$^{ΔuORF}$ mutation. Besides tumour development other parameters contribute to the lifespan extension in female C/EBPβ$^{ΔuORF}$ mice as revealed by the survival curves of the tumour-free female mice (*Figure 3—figure supplement 1E*). Notably, the ageing-associated increase in body weight and body fat was attenuated in female but not in male C/EBPβ$^{ΔuORF}$ mice although at younger age also C/EBPβ$^{ΔuORF}$ males show a reduced body weight and fat content (*Figure 4*). Our earlier data showed that food intake is not reduced in the C/EBPβ$^{ΔuORF}$ mice (*Zidek et al., 2015*) suggesting that the increase in fat catabolism and other features like the observed higher physical activity cause leanness of the C/EBPβ$^{ΔuORF}$ mice (*Zidek et al., 2015*). In accordance with the difference in fat content, we observed a reduction in macrophage infiltration in white adipose tissue from female but not from male C/EBPβ$^{ΔuORF}$ mice (*Figure 4—figure supplement 1D*). Inflammation of the visceral adipose tissue is a common feature of the ageing process and is believed to contribute to insulin resistance and other ageing-associated diseases (*Mau and Yung, 2018*). Therefore, reduced inflammation in adipose tissues could contribute to the extended health and lifespan of the female C/EBPβ$^{ΔuORF}$ mice.

Global liver transcriptome analysis revealed an increase in the inter-individual variation of gene expression between individuals from the same genotype. However, there is less variation between old C/EBPβ$^{ΔuORF}$ females than b e t w e e n old wt females. A similar increase in the inter-individual

variation of gene expression was also identified by others (*Cellerino and Ori, 2017*; *White et al., 2015*) and might reflect different ageing rates within the same group of individuals. Intriguingly, the inter-individual variation in specific pathways and gene groups is different for C/EBPβ^ΔuORF compared to wt mice. Particularly genes connected to metabolic pathways and to ageing-associated diseases showed high expression heterogeneity in old wt but not in old C/EBPβ^ΔuORF females. Whether the increased inter-individual variation of metabolic transcripts in old wt mice is a direct effect of the observed increase of the inhibitory-acting LIP isoform or is due to unknown secondary effects has to be clarified in future studies. It is however conceivable that increased transcriptional robustness in the old C/EBPβ^ΔuORF mice contributes to the extension in health- and lifespan of the female C/EBPβ^ΔuORF mice.

Transcriptome and gene ontology (GO) enrichment analysis in liver revealed some involved mechanisms that could contribute to the youthful and long-lived phenotype of the C/EBPβ^ΔuORF females. We found reduced expression of acute phase response genes in livers from old C/EBPβ^ΔuORF females. Acute phase response genes are associated with inflammation and their expression in the liver increases upon ageing (*Lee et al., 2012*). Moreover, expression of acute phase response genes is inhibited by CR or treatment with the CR-mimetic metformin (*Martin-Montalvo et al., 2013*) suggesting similar protective mechanisms. In addition, we observed the upregulation of several genes connected to lymphocyte biology in the C/EBPβ^ΔuORF livers. This fits to the increase in lymphoplasmatic inflammation in the liver of old female C/EBPβ^ΔuORF mice (*Supplementary file 3*). It is generally believed that ageing associated lymphocyte infiltration rather promotes the ageing process by increasing inflammatory signals (*Singh et al., 2008*) that abrogate glucose homeostasis. Nevertheless, recently this view was challenged by showing that hepatic inflammation, involving the activation of IKKβ, can also be beneficial for maintaining glucose homeostasis (*Liu et al., 2016*). Furthermore, infiltratin g lymphocytes can also contribute to the removal of senescent or pro-tumorigenic cells, thereby acting protective (*Kang et al., 2011*). Further research is required to find out whether in the case of the female C/EBPβ^ΔuORF mice lymphocyte infiltration in the liver has adverse or beneficial effects.

We showed earlier that a *cis*-regulatory uORF in the *Cebpb*-mRNA leader sequence is required for translation into LIP, which is stimulated by mTORC1-4E-BP1 signalling (*Calkhoven et al., 1994*; *Calkhoven et al., 2000*; *Wethmar et al., 2010*; *Zidek et al., 2015*). Intriguingly, other uORF-dependent translation events are known to be involved in lifespan regulation. In yeast, translation of the *GCN4*-mRNA into the GCN4 transcription factor - a basic leucine zipper (bZIP) domain transcription factor like C/EBPβ - is controlled by four uORFs (*Hinnebusch, 2005*). Phosphorylation of the alpha subunit of the eukaryotic initiation factor 2 (eIF2α) by the GCN2 kinase in response to amino acid deprivation or upon other stressors results in global inhibition of translation initiation while GCN4 translation is stimulated due to the skipping of inhibitory uORFs. GCN4 activates genes involved in amino acid biosynthesis and stress response to alleviate nutrient stress (*Hinnebusch, 2005*). GCN4 expression is elevated under different conditions that extend either replicative or chronological lifespan in yeast like glucose restriction, inhibition of TOR signalling, depletion of 60S ribosomal subunits or deletion of the arginine transporter canavanine resistance 1 (CAN1) gene and was shown to be at least partially required for the lifespan extending effects of these interventions (*Beaupere et al., 2017*; *Cherkasova and Hinnebusch, 2003*; *Kubota et al., 2003*; *Martín-Marcos et al., 2007*; *Steffen et al., 2008*; *Valenzuela et al., 2001*; *Yang et al., 2000*). Furthermore, the overexpression of GCN4 is sufficient to extend replicative lifespan in yeast suggesting that GCN4 is a major player in the regulation of yeast lifespan (*Mittal et al., 2017*). In mammals expression of the GCN4 ortholog ATF4 is similarly upregulated in response to stress-induced eIF2α-phosphorylation through skipping of inhibitory uORFs in the *Atf4*-mRNA (*Vattem and Wek, 2004*). Although an involvement of ATF4 in lifespan regulation in mammals has not been addressed so far, increased expression of AT F4 was found in in livers of long-lived mouse models and upon treatments that extend lifespan and in fibroblasts from slow-ageing Snell dwarf and *Pappa* KO mice (*Li et al., 2014*; *Li and Miller, 2015*). In the fibroblasts, increased AT F4 expression was accompanied by an increased stress resistance indicating that AT F4 might play a role also for mammalian lifespan. Notably, C/EBPβ and ATF4 pathways are integrated through heterodimers that bind to composite binding sites (*Fawcett et al., 1999*) suggesting that C/EBPβ-ATF4 dimers are involved in health and lifespan regulation in mammals with C/EBPβ-LAP working together with ATF4 in gene activation while C/EBPβ-LIP probably counteracting it. In yeast the deletion of 60S ribosomal

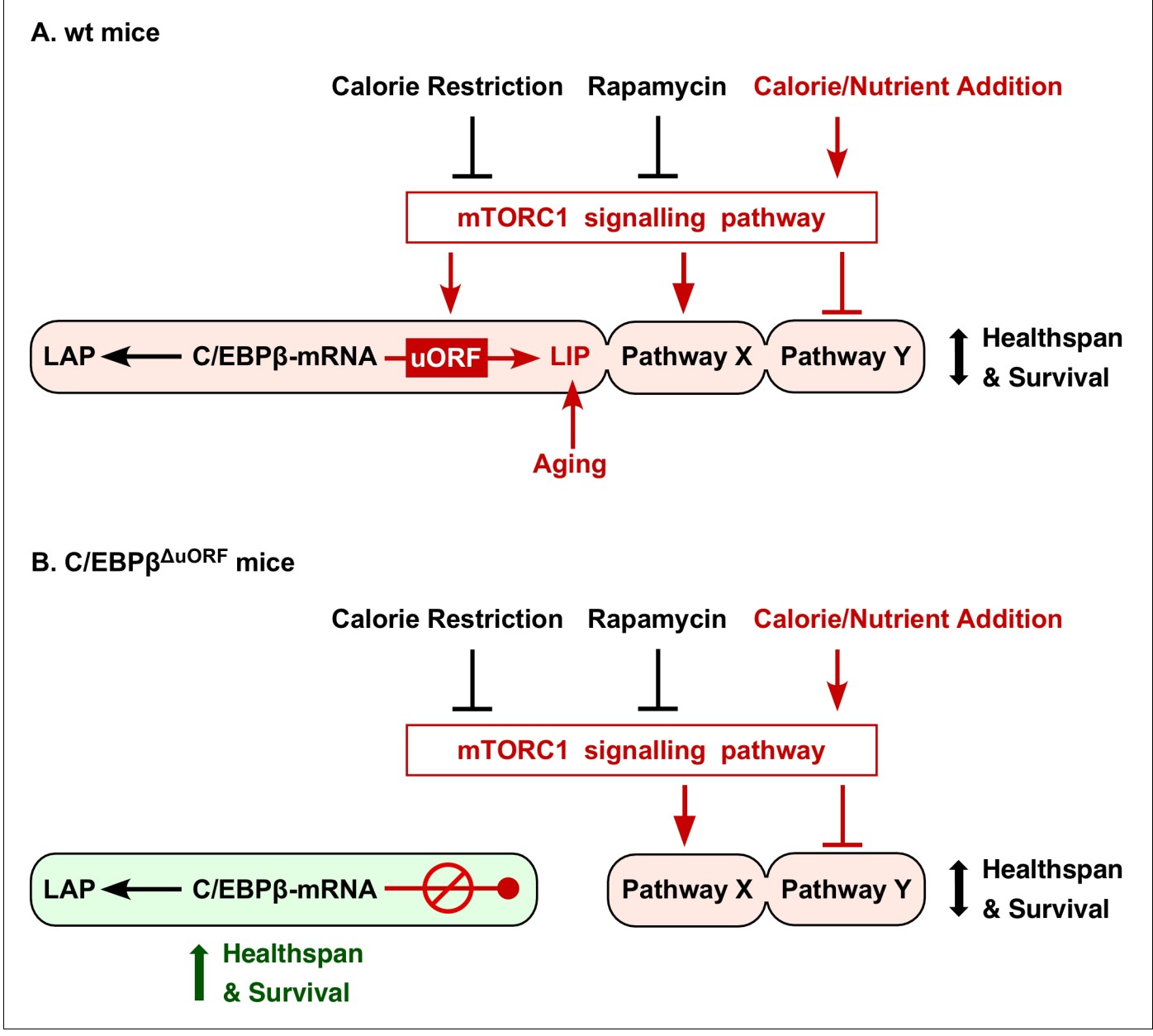

**Figure 8.** Model explaining regulation of LIP under control of mTORC1. (**A**) In wt mice C/EBPβ-mRNA translation into LIP is modulated by calorie/ nutrient availability through mTORC1 signalling, while expression of LAP is not affected. Together with other mTORC1-controlled pathways (Pathway X and Y) LAP/LIP expression ratio determines healthspan and survival. The different pathways may either be, modulated (C/EBPβ), activated (Pathway X) or inhibited (Pathway Y) by mTORC1 and may have different sensitivities to mTORC1 modulators (e.g. rapamycin or nutrients), creating diversity in response (e.g. based on gender, genetic background or age). In addition LIP is upregulated by mechanisms during aging that are not well understood. (**B**) Genetic ablation of the C/EBPβ-uORF prevents the mTORC1-dependent and/or age-associated upregulation of LIP and results in C/EBPβ- dependent health- and lifespan extension. The C/EBPβ$^{\Delta uORF}$ mutation mimics reduced mTORC1 signalling only at the level of LIP expression, leaving mTORC1 control of pathway X and Y unaffected.

DOI: https://doi.org/10.7554/eLife.34985.016

subunits was shown to result in a general reduction of occupancy of uORFs indicating uORF skipping although an effect on translation efficiency of the main reading frame was not observed for most of the mRNAs (*Mittal et al., 2017*). Still there might be a subset of uORF containing mRNAs that might be coregulated under low 60S availability and/or other conditions that result in lifespan extension and mediate the lifespan extending effects. In this respect it is intriguing that uORF-mediated

translation into the C/EBPβ-LIP isoform is reduced upon knockdown or mutation of the Shwachman-Bodian-Diamond Syndrome (SBDS) protein that is required for 60S ribosomal subunit maturation (*In et al., 2016*). Thus, uORF-mediated translation regulation could be a more general mechanism adjusting gene expression during stress response that might play an important role in lifespan extension.

In summary, reduced signalling through the mTORC1 pathway is thought to mediate many of the beneficial effects of CR or rapamycin treatment (*Johnson et al., 2013*), and both conditions restrict mTORC1-controlled translation into LIP (*Calkhoven et al., 2000*; *Zidek et al., 2015*). These and other studies firmly place LIP function downstream of mTORC1 at the nexus of nutrient signalling and metabolic gene regulation (*Figure 8*). However, upon ageing, LIP expression increases (the LAP/LIP ratio decreases) in the liver and WAT whereas significant changes in mTORC1/4E-BP1 signalling were only detected in WAT (*Figure 1—figure supplement 1C–F*). Possibly, in the liver other pathways play a role in age-related upregulation of LIP as has been described for the RNA-binding protein CUGBP1 (*Karagiannides et al., 2001*; *Timchenko et al., 2006*).

Experimental reduction of the transcription factor C/EBPβ-LIP in mice recapitulates many of the effects of CR or treatment with rapamycin, including the reduced cancer incidence and the generally more pronounced extension of lifespan in females. We have developed a high-throughput screening strategy that allows for discovery of small molecular compounds that suppress the translation into LIP (*Zaini et al., 2017*). The identification of such compounds or conditions that reduce LIP translation may reveal new ways of CR-mimetic-based therapeutic strategies beyond those using mTORC1 inhibition.

# Materials and methods

**Key resources table**

| Reagent type (species) or resource | Designation | Source or reference | Identifiers | Additional information |
|---|---|---|---|---|
| Genetic reagent (mus musculus) | C/EBPβ$^{\Delta uORF}$ | DOI: 10.1101/gad. 557910 DOI: 10.15252/embr. 201439837 | NA | mice were further back-crossed to 12 generations into C57BL/6 background |
| Antibody | CD4-PE-Cy7 conjugated | BD Pharmingen | Cat#: 552775 | dilution 1:200 |
| Antibody | CD62L-FITC conjugated | BD Pharmingen | Cat#: 561917 | dilution 1:200 |
| Antibody | CD3e-PE conjugated | eBioscience | Cat#: 12–0031 | dilution 1:200 |
| Antibody | CD8a-eFluor 450 conjugated | eBioscience | Cat#: 48–0081 | dilution 1:200 |
| Antibody | CD44-APC conjugated | eBioscience | Cat#: 17–0441 | dilution 1:200 |
| Antibody | C/EBPβ (E299) (rabbit monoclonal) | Abcam | Cat#: ab32358 | dilution 1:1000 |
| Antibody | β-actin (rabbit polyclonal) | Abcam | Cat#: ab16039 | dilution 1:1000 |
| Antibody | β-actin (clone C4) (mouse monoclonal) | MP Biomedicals | Cat#: 69100 | dilution 1:10000 |
| Antibody | 4E-BP1 (C19) (goat polyclonal) | Santa Cruz | Cat#: sc-6024 | dilution 1:400 |
| Antibody | phospho-4E-BP1 (Thr 37/46) (rabbit polyclonal) | Cell Signaling | Cat#: 9459 | dilution 1:1000 |
| Antibody | p70S6K | Cell Signaling | Cat#: 9202 | dilution 1:1000 |
| Antibody | phospho-p70S6K (Thr389) (108D2) (rabbit monoclonal) | Cell Signaling | Cat#: 9234 | dilution 1:1000 |
| Antibody | S6 ribosomal protein (54D2) (mouse monoclonal) | Cell Signaling | Cat#: 2317 | dilution 1:1000 |
| Antibody | phospho-S6 ribosomal protein (Ser235/236) (2F9) (rabbit monoclonal) | Cell Signaling | Cat#: 4856 | dilution 1:1000 |
| Antibody | HRP-linked anti rabbit IgG | GE Healthcare | Cat#: NA934 | dilution 1:5000 |

*Continued on next page*

*Continued*

| Reagent type (species) or resource | Designation | Source or reference | Identifiers | Additional information |
|---|---|---|---|---|
| Antibody | HRP-linked anti mouse IgG | GE Healthcare | Cat#: NA391 | dilution 1:5000 |
| Antibody | HRP-linked anti goat IgG | Santa Cruz | Cat#: sc-2056 | dilution 1:5000 |
| Sequence-based reagent | *Actb* (β-actin) (F) | DOI: 10.15252/embr.201439837 | NA | 5'-AGAGGGAA ATCGTGCGTGAC-3' |
| Sequence-based reagent | *Actb* (β-actin) (R) | DOI: 10.15252/embr.201439837 | NA | 5'-CAATAGTG ATGACCTGGCCGT-3' |
| Sequence-based reagent | *Cebpb* (F) | DOI: 10.15252/embr.201439837 | NA | 5'-CTGCGGG GTTGTTGAT GT-3' |
| Sequence-based reagent | *Cebpb* (R) | DOI: 10.15252/embr.201439837 | NA | 5'-ATGCTCGA AACGGAAAA GGT-3' |
| Sequence-based reagent | *Cd68* (F) | this paper | NA | 5'-GCCCACC ACCACCAGT CACG-3' |
| Sequence-based reagent | *Cd68* (R) | this paper | NA | 5'-GTGGTCC AGGGTGAGG GCCA-3' |
| Commercial assay or kit | Mouse IGF specific AssayMax ELISA kit | BioCat/Assaypro LLC | Cat#: EMI1001-1-AS | |
| Commercial assay or kit | Lightning Plus ECL reagent | Perkin Elmer | Cat#: NEL103001EA | |
| Commercial assay or kit | Rneasy Plus Mini kit | QIAGEN | Cat#: ID:74134 | |
| Commercial assay or kit | Transcriptor First Strand cDNA Synthesis kit | Roche | Cat#: 4379012001 | |
| Commercial assay or kit | Light Cycler 480 SYBR Green I Master Mix | Roche | Cat#: 04707516001 | |
| Commercial assay or kit | TruSeq Sample Preparation V2 Kit | Illumina | Cat#: RS-122–2002 | |
| Commercial assay or kit | Restore Western Blot Stripping buffer | Thermo Fisher | Cat#: 21063 | |
| Commercial assay or kit | RBC-Lysis buffer | BioLegend | Cat#: 420301 | |
| Commercial assay or kit | QIAzol Lysis re-agent | QIAGEN | Cat#: ID:79306 | |
| Software, algorithm | GraphPad Prism 7.0 | Graphpad Software, La Jolla, CA | | |
| Software, algorithm | DAVID database 6.8 | doi:10.1038/nprot.2008.211 | | |
| Software, algorithm | STAR 2.5.2b | doi:10.1093/bioinformatics/bts635 | | |
| Software, algorithm | Ensembl gene build 86 | http://www.ensembl.org | | |
| Software, algorithm | EdgeR package | doi:10.1152/physiolgenomics.00106.2011 | | |
| Software, algorithm | gProfiler tool | doi:10.1093/nar/gkw199 | | |
| Software, algorithm | Image Quant LAS 4000 Mini Imager software | GE Healthcare | | |

## Mice

C/EBPβ^ΔuORF mice described in (*Wethmar et al., 2010*) were back-crossed for 12 generations into the C57BL/6J genetic background. Mice were kept at a standard 12 hr light/dark cycle at 22°C in individually ventilated cages (IVC) in a specific-pathogen-free (SPF) animal facility on a standard

mouse diet (Harlan Teklad 2916). Mice of the ageing cohort were analysed between 3 and 5 months of age (young) and between 18 and 20 months (old females) or between 20 and 22 months (old males) and were derived from the same breeding pairs as mice used in the lifespan experiment. The body weight of the ageing cohorts was determined before the start of the experimental analysis. All of the animals were handled according to approved institutional animal care and use committee (IACUC) protocols of the Thüringer Landesamt für Verbraucherschutz (#03-005/13) and University of Groningen (#6996A).

## Lifespan experiment

C/EBPβ$^{\Delta uORF}$ and wt littermates (50 mice from each genotype and gender) derived from mating between heterozygous males and females were subjected to a lifespan experiment. Mice were housed in groups with maximum five female mice or four male mice per cage (separated in genotypes and genders) and did not participate in other experiments. Mice were checked daily and the lifespan of every mouse (days) was recorded. Mice were euthanized when the condition of the animal was judged as moribund and/or to be incompatible with continued survival due to severe discomfort based on the independent assessment of experienced animal caretakers. All mice that were found dead or were euthanized underwent necropsy with a few exceptions when the grade of decomposition of dead animals prevented further examination (number of mice without necropsy: n = 0 for wt females; n = 2 for C/EBPβ$^{\Delta uORF}$ females; n = 3 for wt males and n = 5 for C/EBPβ$^{\Delta uORF}$ males. Survival curves were calculated with the Kaplan-Meier method. Statistical significance was determined by the log-rank test using GraphPath Prism 7 software. Maximum lifespan was determined by the number of mice for each genotype that were within the 10% longest-lived mice of the combined (wt and C/EBPβ$^{\Delta uORF}$) cohorts. Statistical significance of observed differences was calculated with Fisher's exact test. In addition, the mean lifespan (± SEM) of the 10% longest lived mice within one genotype was compared to the mean lifespan of the 10% longest lived mice of the other genotype, and the statistical significance was calculated with the Student's T-test.

## Tumour incidence

Suspected tumour tissue found during necropsy of the lifespan cohorts was fixed in 4% paraformaldehyde and Haematoxylin and Eosin stained tissue slices were analysed by experienced board-certified veterinary pathologists of the Dutch Molecular Pathology Centre (Utrecht University) to diagnose the tumour type. Tumour incidence was calculated as percentage of mice with pathologically confirmed tumours in respect to all mice from the same cohort that underwent necropsy. Tumour occurrence was defined as the time of death of an animal in which a pathologically confirmed tumour was found. Tumour load was defined as number of different tumour types found in the same mouse and tumour spread was defined as number of different organs harbouring a tumour within the same mouse irrespective of the tumour type with the exception that in those cases in which different tumour types were found in the same organ a number >1 was rated.

## Motor coordination experiments

Rotarod test: Mice were habituated to the test situation by placing them on a rotarod (Ugo Basile) with constant rotation (5 rpm) for 5 min at two consecutive days with two trials per mouse per day separated by an interval of 30 min. In the test phase, two trials per mouse were performed with accelerating rotation (2–50 rpm within 4 min) with a maximum trial duration of 5 min in which the time was measured until mice fell off the rod. Beam walking test: Mice were trained by using a beam of 3 cm width and 100 cm in length at two consecutive days (one trial per mouse per day). At the test day, mice had to pass a 1 cm wide beam, 100 cm in length and beam crossing time and number off paw slips upon crossing was measured during three trials per mouse that were separated by an interval of 20 min. To determine the number of mistakes the number of paw slips per trial was counted upon examination of recorded videos. Wire Hang test: To measure limb grip strength mice were placed with their four limbs at a grid with wire diameter of 1 mm at 20 cm over the layer of bedding material, and the hanging time was measured until mice loosened their grip and fell down. Three trials of maximal 60 s per mouse were performed that were separated by an interval of 30 min.

## Body composition

The body composition was measured using an Aloka LaTheta Laboratory Computed Tomograph LCT-100A (Zinsser Analytic) as described in (*Zidek et al., 2015*). Percentage body fat was calculated in relation to the sum of lean mass and fat mass.

## Bone measurements

Bones of the hind legs were freed from soft tissue and fixed in 4% paraformaldehyde. For determination of the bone volume, trabecular thickness, trabecular number and trabecular separation femurs were analysed by micro CT (Skyscan 1176, Bruker) equipped with an X-ray tube (50 kV/500µA). The resolution was 9 µm, rotation step was set at 1°C, and a 0.5 mm aluminium filter was used. For reconstruction of femora, the region of interest was defined 0.45 mm (for trabecular bone) or 4.05 mm (for cortical bone) apart from the distal growth plate into the diaphysis spanning 2.7 mm (for trabecular bone) or 1.8 mm (for cortical bone). Trabecular bone volume/tissue volume (%), trabecular number per µm, trabecular thickness (µm) and trabecular separation (intertrabecular distance, µm) was determined according to guidelines by ASBMR Histomorphometry Nomenclature Committee (*Dempster et al., 2013*).

## Glucose tolerance

The intraperitoneal (i.p.) glucose tolerance test (IPGTT) was performed as described in (*Zidek et al., 2015*). Mice without initial increase in blood glucose concentration were excluded from the analysis.

## Flow cytometry

Blood cells from 300 µl blood were incubated in RBC-Lysis buffer (Biolegend) to lyse the red blood cells. Remaining cells were washed and incubated with a cocktail of fluorochrome-conjugated antibodies (Cd4-PE-Cy7 (#552775) and Cd62L-FITC (#561917) from BD Pharmingen; Cd3e-PE (#12–0031), Cd8a-eFluor 450 (#48–0081) and Cd44-APC (#17–0441) from eBioscience.), incubated with propidium iodide for the detection of dead cells and analysed using the FACSCanto II analyser (BD Biosciences). The following T cell subsets were quantified: $Cd3^+$, $Cd8^+$, $Cd44^{high}$ cytotoxic memory T cells; $Cd3^+$, $Cd8^+$, $Cd44^{low}$, $Cd62L^{high}$ cytotoxic naïve T cells, $Cd3^+$, $Cd4^+$, $Cd44^{high}$ helper memory T cells and $Cd3^+$, $Cd4^+$, $Cd44^{low}$, $Cd62L^{high}$ helper naïve T cells.

## Histology

Tissue pieces were fixed with 4% paraformaldehyde and embedded in paraffin. Sections were stained with Haematoxylin and Eosin (H and E) and agei n g-related pathologies or tumour types were analysed by experienced board-certified veterinary pathologists of the Dutch Molecular Pathology Centre (Utrecht University). Semi-quantification of muscle regeneration was done by counting the number of myofibers with a row of internalized nuclei (>4) for five 200x fields. Other ageing-associated lesions were scored subjectively, and the severity of the lesions was graded on a scale between 0 and 3 with 0 = absent; 1 = mild; 2 = moderate and 3 = severe.

## Immunoblotting and quantification

Mouse liver and WAT tissue was homogenized on ice with a glass douncer in RIPA buffer (150 mM NaCl, 1% NP40, 0.5% sodium deoxycholate, 0.1% SDS, 50 mM TRIS pH 8.0 supplemented with protease and phosphatase inhibitors). Liver extracts were sonicated immediately, WAT extracts were incubated for 1 hr on ice, centrifuged for 15 min at 4°C after which the lipid layer was carefully removed using a cotton bud and then sonicated. Equal amounts of total protein were separated by SDS-PAGE, transferred to a PVDF membrane and incubated with the following antibodies: C/EBPβ (E299) from Abcam, β-actin (ab16039) from Abcam or (# 69100, clone C4) from MP Biomedicals; 4E-BP1 (C-19) from Santa Cruz; phospho-p70S6K (Thr389) (108D2), p70S6K (#9202), phospho-S6 ribosomal protein (Ser235/236) (2F9), S6 ribosomal protein (54D2), and phospho-4E-BP1 (Thr 37/46) (#9459) from Cell Signaling Technology and HRP-linked anti rabbit or mouse IgG from GE Healthcare and HRP-linked anti goat IgG from Santa Cruz. Lightning Plus ECL reagent (Perkin Elmer) was used for detection and for re-probing membranes were incubated in Restore Western Blot Stripping buffer (Thermo Fisher). The detection and quantification of protein bands was performed with the Image Quant LAS 4000 Mini Imager (GE Healthcare) using the supplied software.

## Quantitative real-time PCR

Mouse liver or visceral fat tissue was homogenized on ice with a motor driven pellet pestle (Kontes) in the presence of QIAzol reagent (QIAGEN) and total RNA was isolated as described in (*Zidek et al., 2015*). cDNA synthesis was performed from 1 µg of total RNA with the Transcriptor First Strand cDNA Synthesis Kit (Roche) using random hexamer primers. qRT-was performed with the LightCycler 480 SYBR Green I Master mix (Roche) using the following primers: *Actb* (β-actin): 5'-AGA GGG AAA TCG TGC GTG AC-3' and 5'-CAA TAG TGA TGA CCT GGC CGT-3'; *Cebpb*: 5'-CTG CGG GGT TGT TGA TGT-3' and 5'-ATG CTC GAA ACG GAA AAG GT-3'; *Cd68*: 5'-GCC CAC CAC CAC CAG TCA CG-3' and 5'- GTG GTC CAG GGT GAG GGC CA-3'.

## Enzyme-linked immunosorbent assay (ELISA)

Plasma was prepared as described in (*Zidek et al., 2015*) and the IGF-1 specific ELISA was performed according to the instructions of the manufacturer (BioCat).

## RNA-seq analysis

Liver tissue from young (5 months) and old (20 months) wt and C/EBPβ$^{ΔuORF}$ mice (from six individuals per group) was homogenized on ice with a motor-driven pellet pestle (Kontes) in the presence of QIAzol reagent (Qiagen), and total RNA was isolated as described in (*Zidek et al., 2015*). Preparation of the sequencing libraries was performed using the TruSeq Sample Preparation V2 Kit (Illumina) according to the manufacturer's instructions. High-throughput single-end sequencing (65 bp) of the libraries was performed with an Illumina HiSeq 2500 instrument. Reads were aligned and quantified using STAR 2.5.2b (*Dobin et al., 2013*) against primary assembly GRCm38 using Ensembl gene build 86 (http://www.ensembl.org). Genes with average expression level below one fragment per million (FPM) were excluded from the analysis. A generalized linear model was used to identify differential gene expression using EdgeR package (*McCarthy et al., 2012*; *Robinson et al., 2010*). The library normalization was left at the standard setting (trimmed mean of M-values, TMM). The resulting p-values were corrected for multiple testing using the Benjamini-Hochberg procedure. Data visualization, calculation of CV (coefficient of variation) and statistical tests were conducted using custom R scripts (Processed data and R script available at http://www.genomes.nl/CEBPB_delta_uORF/ [*de Jong and Guryev, 2018b*] or https://github.com/Vityay/CEBPB_delta_uORF [(*de Jong and Guryev, 2018a*]; copy archived at https://github.com/elifesciences-publications/CEBPB_delta_uORF). Gene ontology (GO) analysis was performed using the DAVID database version 6.8 (*Huang et al., 2009*) with default DAVID database setting with medium stringency and *Mus musculus* background. KEGG pathway analysis was performed using gProfiler tool (*Reimand et al., 2016*). For dataset see (*Müller et al., 2018*).

## Statistical analysis

Biological replication is indicated (n = x). All graphs show average ± standard error of the mean (s.e.m.). The unpaired, two-tailed Student's t-Test was used to calculate statistical significance of results with *p<0.05; **p<0.01; ***p<0.001. Significance of the differences in survival curves was analysed using the log-rank test using Prism7 (GraphPad Software) and significance of the difference in maximum lifespan (number of mice from one cohort within the 10% longest lived mice calculated from the combined cohort) and tumour incidence was calculated using the Fisher's exact test with *p<0.05. Daily Chi-square test calculations were carried out to examine the significance of parts of the survival curves.

## Acknowledgements

We thank Rafael de Cabo, NIA Baltimore for advice with the lifespan experiment. At the ERIBA/UMCG Groningen, we thank Mirjam Koster for technical assistance with histology and Gerald de Haan for critical reading of the manuscript. At the FLI, we thank we thank Sabrina Eichwald for technical assistance, Lucien Frappart and Dominique Galendo for advice on necropsy, the staff of the animal house facility in particular Anja Baar and Juliane Brüchert for massive support with the lifespan experiment, Nico Andreas for advice concerning the flow cytometry experiments, Anne Gompf for technical assistance with flow cytometry, Maik Baldauf for paraffin embedding and Christina Valkova

for advice with the motor coordination experiments. LMZ was supported by the Deutsche Forschungsgemeinschaft (DFG) through a grant to CFC. (CA 283/1–1) and a grant to JVM. (MA-3975/2–1). TA. was supported by the Leibniz Graduate School on Ageing and Age-Related Diseases (LGSA; www.fli-leibniz.de/phd/) PL. was supported by the Collaborative Research Centre 1149 'Trauma' (INST 40/492–1) and DFG priority program Immunobone (Tu220/6) to a grant to JPT.

## Additional information

### Funding

| Funder | Grant reference number | Author |
| --- | --- | --- |
| Deutsche Forschungsgemeinschaft | CA 283/1-1 | Laura M Zidek<br>Cornelis F Calkhoven |
| Leibniz-Gemeinschaft | LGSA | Tobias Ackermann<br>Cornelis F Calkhoven |
| Deutsche Forschungsgemeinschaft | MA 3975/2-1 | Laura M Zidek<br>Julia von Maltzahn |
| Deutsche Forschungsgemeinschaft | INST 40/492-1 | Peng Liu<br>Jan P Tuckermann |
| Deutsche Forschungsgemeinschaft | TU 220/6 | Peng Liu<br>Jan P Tuckermann |

The funders had no role in study design, data collection and interpretation, or the decision to submit the work for publication.

### Author contributions

Christine Müller, Laura M Zidek, Conceptualization, Data curation, Formal analysis, Supervision, Validation, Investigation, Visualization, Methodology, Writing—original draft, Project administration, Writing—review and editing; Tobias Ackermann, Formal analysis, Validation, Investigation, Methodology, Writing—review and editing; Tristan de Jong, Data curation, Formal analysis, Validation, Investigation, Visualization, Methodology, Writing—review and editing; Peng Liu, Gertrud Kortman, Formal analysis, Validation, Investigation, Methodology; Verena Kliche, Formal analysis, Investigation, Methodology; Mohamad Amr Zaini, Formal analysis, Validation, Investigation; Liesbeth Harkema, Formal analysis, Validation, Investigation, Visualization; Dineke S Verbeek, Formal analysis, Investigation; Jan P Tuckermann, Resources, Supervision, Methodology; Julia von Maltzahn, Alain de Bruin, Supervision, Validation, Investigation, Methodology, Writing—review and editing; Victor Guryev, Data curation, Formal analysis, Supervision, Investigation, Visualization, Methodology, Writing—review and editing; Zhao-Qi Wang, Conceptualization, Resources, Supervision, Methodology, Writing—review and editing; Cornelis F Calkhoven, Conceptualization, Formal analysis, Supervision, Funding acquisition, Validation, Visualization, Methodology, Writing—original draft, Project administration, Writing—review and editing

### Author ORCIDs

Christine Müller (iD) http://orcid.org/0000-0003-1974-4053
Cornelis F Calkhoven (iD) http://orcid.org/0000-0001-6318-7210

### Ethics

Animal experimentation: All of the animals were handled according to approved institutional animal care and use committee (IACUC) protocols of the Thüringer Landesamt für Verbraucherschutz (#03-005/13) and University of Groningen (#6996A).

### Decision letter and Author response

Decision letter https://doi.org/10.7554/eLife.34985.028
Author response https://doi.org/10.7554/eLife.34985.029

## Additional files

### Supplementary files

• Supplementary file 1. Table 1 Lifespan experiment summary of results. [1] Number of animals in the cohort [2] Median survival of the cohort (days) [3] Increase of the median survival (percent) [4] P-value of the increased survival (log-rank test) [5] Mean lifespan of the cohort (days) [6] Standard error of the mean [7] Number of mice in the cohort in the longest-lived decile of the combined cohort (wt and C/EBPβ$^{\Delta uORF}$) [8] P-value of increased $N_{max}$[7] (Fisher's exact test).

DOI: https://doi.org/10.7554/eLife.34985.017

• Supplementary file 2. Table 2 Tumour spectrum in wt and C/EBPβ$^{\Delta uORF}$ mice. *incl. lymphoid leukaemia **incl. malignant round cell neoplasms ***tumour type could not be unequivocally determined due to inadequate quality of the fixed tumour tissue Absolute numbers of mice with the indicated tumour type of tumours found during necropsy for wt and C/EBPβ$^{\Delta uORF}$ males and females are shown. Note that the total number of tumours is higher than the number of tumour-bearing mice due to the eventual occurrence of different tumour types in the same mouse.

DOI: https://doi.org/10.7554/eLife.34985.018

• Supplementary file 3. Table 3 Occurrence of ageing-associated pathologies in wt and C/EBPβ$^{\Delta uORF}$ mice. Mice were part of the ageing cohort and the age at analysis was 20 months for females and 22 months for males. [1] Number of animals showing the pathology (out of the total number of animals analyzed). [2] mean grade of the pathology as calculated from the total number of animals analyzed with 0 = absent, 1 = mild, 2 = moderate and 3 = severe. [3] Statistical significance of difference found between wt and C/EBPβ$^{\Delta uORF}$ mice from the same gender as calculated using the Student's t-test (ns = not significant). [4] Mean number of regenerating muscle fibers found in five histological tissue slices per mouse. Note that a lower number is an indication for a more progressed ageing phenotype. [5] mean surface area of intramuscular adipose tissue in percent of the total area of analyzed skeletal muscle tissue as calculated from the total number of animals analyzed. [6] Trabecular bone parameters (percent bone volume/tissue volume; Trabecular number per mm; trabecular thickness and trabecular separation) measured by micro-CT analysis.

DOI: https://doi.org/10.7554/eLife.34985.019

• Supplementary file 4. Table 4 GO-term analysis of genes upregulated in livers of old C/EBPβ$^{\Delta uORF}$ mice. Functional annotation of genes upregulated in livers of old C/EBPβ$^{\Delta uORF}$ female mice compared to livers of old wt female mice (FDR < 0.01; 103 from 127 genes; 24 unknown IDs) using the DAVID database (*Huang et al., 2009*).

DOI: https://doi.org/10.7554/eLife.34985.020

• Supplementary file 5. Table 5 GO-term analysis of genes downregulated in livers of old C/EBPβ$^{\Delta uORF}$ mice. Functional annotation of genes downregulated in livers of old C/EBPβ$^{\Delta uORF}$ female mice compared to livers of old wt female mice (FDR < 0.01; 23 from 25 genes, two unknown IDs) using the DAVID database (*Huang et al., 2009*).

DOI: https://doi.org/10.7554/eLife.34985.021

• Supplementary file 6. Table 6 GO-term analysis of genes showing high inter-individual variation between livers of old wt female mice. Functional annotation of genes showing a high inter-individual variation between livers from old wt female mice. Coefficient of variation of transcript levels of the corresponding gene in the livers of wt mice is at least twice as big as the coefficient of variation of the transcript levels of the same gene in the livers from C/EBPβ$^{\Delta uORF}$ mice; 1386 from 1414 genes, 28 unknown IDs; using the DAVID database (*Huang et al., 2009*).

DOI: https://doi.org/10.7554/eLife.34985.022

• Supplementary file 7. Table 7 GO-term analysis of genes showing high inter-individual variation between livers of old C/EBPβ$^{\Delta uORF}$ female mice. Functional annotation of genes showing a high inter-individual variation between livers from old wt female mice. Coefficient of variation of transcript levels of the corresponding gene in the livers of C/EBPβ$^{\Delta uORF}$ mice is at least twice as big as the coefficient of variation of the transcript levels of the same gene in the livers from wt mice; 1354 from 1375 genes, 21 unknown IDs; using the DAVID database (*Huang et al., 2009*).

DOI: https://doi.org/10.7554/eLife.34985.023

• Transparent reporting form

eLIFE Research article

DOI: https://doi.org/10.7554/eLife.34985.024

### Data availability

For transcriptome dataset see: Müller, C., de Jong, T, Guryev, V, Calkhoven, CF. (2018). Transcriptome profiling of liver samples of C/EBPβΔuORF mice. Retrieved from: https://www.ebi.ac.uk/arrayexpress/

The following dataset was generated:

| Author(s) | Year | Dataset title | Dataset URL | Database, license, and accessibility information |
|---|---|---|---|---|
| Müller C, de Jong T, Guryev V and Calkhoven CF | 2018 | Transcriptome profiling of liver samples of C/EBP$\beta$ΔuORF mice | https://www.ebi.ac.uk/arrayexpress/experiments/E-MTAB-6435/ | Publicly available at the Electron Microscopy Data Bank (accession no. E-MTAB-6435) |

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
