## [Decision Letter]

Thank you for submitting your article "Reduced expression of C/EBPβ-LIP extends health- and lifespan in mice" for consideration by *eLife*. Your article has been reviewed by three peer reviewers, and the evaluation has been overseen by a Reviewing Editor and James Manley as the Senior Editor. The following individuals involved in review of your submission have agreed to reveal their identity: Z. David Sharp (Reviewer #1); Matt Kaeberlein (Reviewer #2).

The reviewers have discussed the reviews with one another and the Reviewing Editor has drafted this decision to help you prepare a revised submission.

Summary:

The authors examined the phenotypes of a mouse with a deletion in the uORF of C/EBPβ. They find that female mice live longer and have several attenuated phenotypes of aging.

This is a well-written and important study that convincingly links CR and rapamycin-mediated lifespan extension to C/EBPβ regulation of LAP/LIP.

Essential revisions:

There is a general consensus among the three reviewers that this is an interesting paper and in principle could be suitable for publication in *eLife*. However, two major issues need be addressed prior to publication.

1) The dose/sex response requires more discussion. With respect to dose, the data from the ITP clearly suggest that males (at least in UMHET3) don't see as much benefit from rapamycin at low doses in terms of lifespan, but they "catch up" to females at higher doses. Thus, the statement that effects "are much stronger in females" is a bit misleading, as the only dose tested starting in mid-life is the low 14 ppm (1X ITP) dose. Bitto et al., by studying transient rapamycin treatment, saw that at the higher dose males but not females showed a lifespan extension. However, at the lower dose tested (9X ITP), both males and females had their lifespans extended, pretty comparably, but with males perhaps getting a little more benefit. Obviously, the mechanisms behind these effects are not completely known (although the current study is highly suggestive), but one possibility is that rapamycin is doing at least two things: affecting LAP/LIP and something else. Perhaps LAP/LIP is more sensitive than pathway #2 to mTORC1 inhibition and so at lower doses of rapamycin this happens first, which is why females get more of a benefit at lower dose. At higher doses, pathway #2 is engaged and both males and females get the benefit. These different possible mechanisms should be discussed.

2) Information about LAP/LIP changes with age would be very useful, and whether it is correlated with changes in mTOR signaling. If the authors can perform this with banked tissue it would be an excellent experiment.

3) Another issue the relatively short lifespan of the control population of mice. Even excluding the mice that were sacrificed due to ulcerative dermatitis, the control population is borderline short-lived. Interesting both the UD and non-UD females seem to be extended in the female mutant mice and, although with the range of other ageing phenotypes examined, the concerns of the control population are somewhat mitigated.

---

## [Author Response]

Summary:The authors examined the phenotypes of a mouse with a deletion in the uORF of C/EBPβ. They find that female mice live longer and have several attenuated phenotypes of aging.This is a well-written and important study that convincingly links CR and rapamycin-mediated lifespan extension to C/EBPβ regulation of LAP/LIP.

The paper now contains separate Results and Discussion sections to improve the flow (Discussion section). The Discussion section has become quite large. We hope that you agree that because of the many different aspects of the study this is justified.

We have addressed all your questions/comments and you will find our replies below.

Essential revisions:There is a general consensus among the three reviewers that this is an interesting paper and in principle could be suitable for publication in eLife. However, two major issues need be addressed prior to publication.1) The dose/sex response requires more discussion. With respect to dose, the data from the ITP clearly suggest that males (at least in UMHET3) don't see as much benefit from rapamycin at low doses in terms of lifespan, but they "catch up" to females at higher doses. Thus, the statement that effects "are much stronger in females" is a bit misleading, as the only dose tested starting in mid-life is the low 14 ppm (1X ITP) dose. Bitto et al., by studying transient rapamycin treatment, saw that at the higher dose males but not females showed a lifespan extension. However, at the lower dose tested (9X ITP), both males and females had their lifespans extended, pretty comparably, but with males perhaps getting a little more benefit. Obviously, the mechanisms behind these effects are not completely known (although the current study is highly suggestive), but one possibility is that rapamycin is doing at least two things: affecting LAP/LIP and something else. Perhaps LAP/LIP is more sensitive than pathway #2 to mTORC1 inhibition and so at lower doses of rapamycin this happens first, which is why females get more of a benefit at lower dose. At higher doses, pathway #2 is engaged and both males and females get the benefit. These different possible mechanisms should be discussed.

We now discuss these issues in the Discussion session. We also discuss now the possible involvement of C/EBP-hormone receptor regulation as well as possible effects of higher LAP expression in C/EBPβ^ΔuORF^ males (Discussion section).

2) Information about LAP/LIP changes with age would be very useful, and whether it is correlated with changes in mTOR signaling. If the authors can perform this with banked tissue it would be an excellent experiment.

We may have not been clear enough in our description, but the data (as far as we understand the question well) are already in the paper: Figure 1B,C shows immunoblots from livers of young and old mice for both wt and ΔuORF mutation with LAP/LIP ratio quantifications at the right. In old wt mice LAP/LIP ratios decrease through upregulation of LIP as we describe in the Results section. We now added data showing LIP upregulation in WAT of old females (Figure 1—figure supplement 1B). Unfortunately, we lost WAT stored from males during transport to my new institute.

Figure 1—figure supplement 1C-F shows that in liver we did not find significant changes in the mTORC1-downstream P-S6K/S6K or P-4E-BP1/4E-BP1 between young and old mice. In female WAT P-S6/S6 does not change between young and old, but P-4E-BP1/4E-BP1 does increase in old for both wt and C/EBPβ^ΔuORF^. Therefore, the interpretation is not so easy: we cannot relate these changes in LAP/LIP to changes in mTORC1 signalling with age in liver and in WAT only the P-4E-BP1/4E-BP1 is increased but not P-S6/S6. See Discussion section.

3) Another issue the relatively short lifespan of the control population of mice. Even excluding the mice that were sacrificed due to ulcerative dermatitis, the control population is borderline short-lived. Interesting both the UD and non-UD females seem to be extended in the female mutant mice and, although with the range of other ageing phenotypes examined, the concerns of the control population are somewhat mitigated.

We consider that this is a comment(?) without requiring our action.